# JAK inhibition decreases the autoimmune burden in Down syndrome

Angela L Rachubinski[1,2]*, Elizabeth Wallace[3], Emily Gurnee[3], Belinda A Enriquez-Estrada[1], Kayleigh R Worek[1], Keith P Smith[1], Paula Araya[1], Katherine A Waugh[1†], Ross E Granrath[1], Eleanor Britton[1], Hannah R Lyford[1], Micah G Donovan[1,4], Neetha Paul Eduthan[1], Amanda A Hill[1], Barry Martin[5], Kelly D Sullivan[1,6], Lina Patel[1,7], Deborah J Fidler[8], Matthew D Galbraith[1,4], Cory A Dunnick[3], David A Norris[3], Joaquín M Espinosa[1,4]*

[1]Linda Crnic Institute for Down Syndrome, University of Colorado Anschutz Medical Campus, Aurora, United States; [2]Department of Pediatrics, Section of Developmental Pediatrics, University of Colorado Anschutz Medical Campus, Aurora, United States; [3]Department of Dermatology, University of Colorado Anschutz Medical Campus, Aurora, United States; [4]Department of Pharmacology, University of Colorado Anschutz Medical Campus, Aurora, United States; [5]Department of Internal Medicine, University of Colorado Anschutz Medical Campus, Aurora, United States; [6]Department of Pediatrics, Section of Developmental Biology, University of Colorado Anschutz Medical Campus, Aurora, United States; [7]Department of Psychiatry, Child and Adolescent Division, University of Colorado Anschutz Medical Campus, Aurora, United States; [8]Department of Human Development and Family Studies, Colorado State University, Fort Collins, United States

*For correspondence:
angela.rachubinski@cuanschutz.
edu (ALR);
joaquin.espinosa@cuanschutz.
edu (JME)

Present address: †Department of Cell Biology and Physiology, University of Kansas Medical Center, Kansas City, United States

## eLife Assessment

Rachubinski and colleagues provide an **important** manuscript that includes two major advances in understanding immune dysregulation in a large cohort of individuals with Down syndrome. The work comprises **compelling**, comprehensive, and state-of-the-art clinical, immunological, and auto-antibody assessment of autoimmune/inflammatory manifestations. Additionally, the authors report promising results from a clinical trial with the JAK inhibitor tofacitinib for individuals with dermatological autoimmune disease.

## Abstract

**Background:** Individuals with Down syndrome (DS), the genetic condition caused by trisomy 21 (T21), display clear signs of immune dysregulation, including high rates of autoimmunity and severe complications from infections. Although it is well established that T21 causes increased interferon responses and JAK/STAT signaling, elevated autoantibodies, global immune remodeling, and hyper-cytokinemia, the interplay between these processes, the clinical manifestations of DS, and potential therapeutic interventions remain ill defined.

**Methods:** We report a comprehensive analysis of immune dysregulation at the clinical, cellular, and molecular level in hundreds of individuals with DS, including autoantibody profiling, cytokine analysis, and deep immune mapping. We also report the interim analysis of a Phase II clinical trial investigating the safety and efficacy of the JAK inhibitor tofacitinib through multiple clinical and molecular endpoints.

**Results:** We demonstrate multi-organ autoimmunity of pediatric onset concurrent with unexpected autoantibody-phenotype associations in DS. Importantly, constitutive immune remodeling and hypercytokinemia occur from an early age prior to autoimmune diagnoses or autoantibody production. Analysis of the first 10 participants to complete 16 weeks of tofacitinib treatment shows a good safety profile and no serious adverse events. Treatment reduced skin pathology in alopecia areata, psoriasis, and atopic dermatitis, while decreasing interferon scores, cytokine scores, and levels of pathogenic autoantibodies without overt immune suppression.

**Conclusions:** JAK inhibition is a valid strategy to treat autoimmune conditions in DS. Additional research is needed to define the effects of JAK inhibition on the broader developmental and clinical hallmarks of DS.

**Funding:** NIAMS, Global Down Syndrome Foundation.

**Clinical trial number:** NCT04246372.

## Introduction

Trisomy of human chromosome 21 (T21) occurs at a rate of ~1 in 700 live births, causing Down syndrome (DS; *Lejeune et al., 1959*; *Antonarakis et al., 2020*). Individuals with DS display a distinct clinical profile including developmental delays, stunted growth, cognitive impairments, and increased risk of leukemia, autism spectrum disorders, seizure disorders, and Alzheimer's disease (*Antonarakis et al., 2020*; *Chicoine et al., 2021*). People with DS also display widespread immune dysregulation, which manifests through severe complications from respiratory viral infections and high prevalence of myriad immune conditions, including autoimmune thyroid disease (AITD; *Iughetti et al., 2014*; *Pierce et al., 2017*; *Amr, 2018*), celiac disease (*Zachor et al., 2000*; *Book et al., 2001*), and skin conditions such as atopic dermatitis, alopecia areata, hidradenitis suppurativa (HS), vitiligo, and psoriasis (*Madan et al., 2006*; *Sureshbabu et al., 2011*; *Lam et al., 2020*). Furthermore, people with DS display signs of neuroinflammation from an early age (*Wilcock and Griffin, 2013*; *Flores-Aguilar et al., 2020*; *Araya et al., 2022*). Although it is now well accepted that immune dysregulation is a hallmark of DS, the underlying mechanisms and therapeutic implications are not yet fully defined.

We previously reported that T21 causes consistent activation of the interferon (IFN) transcriptional response in multiple immune and non-immune cell types with concurrent hypersensitivity to IFN stimulation and hyperactivation of downstream JAK/STAT signaling (*Sullivan et al., 2016*; *Waugh et al., 2019*; *Araya et al., 2019*). Plasma proteomics studies identified dozens of inflammatory cytokines with mechanistic links to IFN signaling that are elevated in people with DS (*Sullivan et al., 2017*). A large metabolomics study revealed that T21 drives the production of neurotoxic tryptophan catabolites via the IFN-inducible kynurenine pathway (*Powers et al., 2019*). Deep immune profiling revealed global immune remodeling with hypersensitivity to IFN across all major branches of the immune system (*Waugh et al., 2019*), and dysregulation of T cell lineages toward a hyperactive, autoimmunity-prone state (*Araya et al., 2019*). These results could be partly explained by the fact that four of the six IFN receptors (IFNRs) are encoded on chr21, including Type I, II, and III IFNR subunits (*Secombes and Zou, 2017*). In a mouse model of DS, normalization of *IFNR* gene copy number rescues multiple phenotypes of DS, including lethal immune hypersensitivity, congenital heart defects (CHDs), cognitive impairments, and craniofacial anomalies (*Waugh et al., 2023*). JAK inhibition rescues lethal immune hypersensitivity in these mouse models (*Tuttle et al., 2020*) and attenuates the global dysregulation of gene expression caused by the trisomy across multiple murine tissues (*Galbraith et al., 2023*). Furthermore, prenatal JAK inhibition in pregnant mice prevents the appearance of CHDs (*Chi et al., 2023*). Altogether, these results support the notion that T21 elicits an interferonopathy in DS, and that pharmacological inhibition of IFN signaling could have multiple therapeutic benefits in this population.

Although it is now well established that T21 disrupts immune homeostasis toward an autoimmunity-prone state, the interplay between overexpression of chromosome 21 genes, hyperactive interferon signaling, dysregulation of immune cell lineages, autoantibody production, hypercytokinemia, and the various developmental and clinical features of DS remain to be elucidated. Previous studies established similarities between the immune profiles of typical aging, autoimmunity in the general population, and DS, proposing a role for accelerated immune aging in the pathophysiology of DS (*Gensous et al., 2020*; *Lambert et al., 2022*; *Khor and Buckner, 2023*). Other studies indicate a role for

elevated cytokine production, hyperactivated T cells, and ongoing B cell activation as drivers of auto-immunity in DS (*Waugh et al., 2019*; *Araya et al., 2019*; *Malle et al., 2023*). However, given the relatively small sample sizes and observational nature of these studies, it has not been possible to define the contribution of specific dysregulated events to breach of tolerance leading to clinically evident autoimmunity in DS. Therefore, additional research is needed to define driver versus bystander events that could illuminate therapeutic strategies to decrease the burden of autoimmunity in DS.

Within this context, we report here a comprehensive analysis of the immune disorder of DS, including detailed annotation of autoimmune and inflammatory conditions and quantification of auto-antibodies in hundreds of research participants, which reveals widespread autoimmune attack on all major organ systems in DS from an early age, including unexpected autoantibody-phenotype associations. Then, using deep immune mapping and quantitative proteomics, we demonstrate that T21 causes widespread immune remodeling toward an autoimmunity-prone state accompanied by hyper-cytokinemia prior to clinically evident autoimmunity or autoantibody production. Lastly, we report the interim analysis of a clinical trial investigating the safety and efficacy of the JAK1/3 inhibitor tofacitinib (Xeljanz, Pfizer) in DS. These results demonstrate that JAK inhibition improves multiple immunodermatological conditions in DS, normalizes interferon scores, decreases levels of major pathogenic cytokines (e.g. TNF-α, IL-6), and reduces levels of pathogenic autoantibodies (e.g. anti-thyroid peroxidase [anti-TPO]). Altogether, these results point to hyperactive JAK/STAT signaling as driver of autoimmunity in DS and justify the ongoing trials of JAK inhibitors in DS for multiple clinical endpoints.

## Methods

### Human Trisome Project (HTP) study

All aspects of this study were conducted in accordance with the Declaration of Helsinki under protocols approved by the Colorado Multiple Institutional Review Board. Results and analyses presented herein are part of a nested study within the Crnic Institute's Human Trisome Project (HTP, NCT02864108, see also https://www.trisome.org/) cohort study. All study participants, or their guardian/legally authorized representative, provided written informed consent. The HTP study has generated multiple multi-omics datasets on hundreds of research participants, some of which have been analyzed in previous studies, including whole blood transcriptome data (*Galbraith et al., 2023*; *Donovan et al., 2024a*; *Donovan et al., 2024b*), white blood cell transcriptome data (*Powers et al., 2019*), plasma proteomics (*Galbraith et al., 2023*), plasma metabolomics (*Powers et al., 2019*; *Galbraith et al., 2023*), and immune mapping via flow cytometry (*Araya et al., 2019*) and mass cytometry (*Galbraith et al., 2023*; *Waugh et al., 2023*). This paper reports new analyses of select previous datasets (transcriptome, mass cytometry, MSD immune markers) within the larger multi-omics dataset of the HTP study, as well as analyses of new datasets (e.g. anti-TPO, ANA, autoantibodies), as described in detail below.

### Annotation of co-occurring conditions

Within the HTP, a clinical history for each participant is curated from both medical records and participant/family reports. Both surveys are set up as REDCap (*Harris et al., 2009*) instruments that collect information as a review of systems (e.g. cardiovascular, immunity, endocrine). Expert data curators complete the medical record review and evaluate answers provided by self-advocates and caregivers. In cases of discordant answers across the two instruments, medical records take precedence. De-identified demographic and clinical metadata obtained is then linked to de-identified biospecimens used to generate the various -omics (e.g. RNA sequencing) and targeted assay datasets (e.g. anti-TPO assays). For annotation of AITD, several possible entries were considered as shown in *Figure 1—figure supplement 1a*, including history of hypothyroidism, hyperthyroidism, Hashimoto's disease, Grave's disease, anti-TPO or -TG antibodies, and subclinical hypothyroidism. For annotation of immune skin conditions, atopic dermatitis and eczema were combined and counted in a single group, as were hidradenitis suppurativa (HS), folliculitis, and 'boils'.

### Blood sample collection and processing

The biological datasets analyzed herein were derived from peripheral blood samples collected using PAXgene RNA Tubes (Qiagen) and BD Vacutainer K2 EDTA tubes (BD). Whole blood from PAXgene

collection tubes was processed for RNA sequencing as described below. Two 0.5 mL aliquots of whole blood were withdrawn from each EDTA tube and processed for mass cytometry as described below. The remaining EDTA blood samples were centrifuged at 700 x *g* for 15 min to separate plasma, buffy coat containing white blood cells (WBC), and red blood cells (RBCs). Samples were then aliquoted, flash frozen and stored at –80 °C until subsequent processing and analysis. Centrifugation and storage of samples took place within 2 hr of collection.

## Measurements of autoantibodies

Anti-TPO status was determined from plasma samples using an electrochemiluminescence-based assay (*Gu et al., 2019*), and carried out by the Autoantibody/HLA Core Facility of the Barbara Davis Center for Childhood Diabetes at the University of Colorado Anschutz Medical Campus. Sample values were calculated as (sample signal – negative control signal) / (positive control signal – negative control signal), with the threshold (upper limit of normal) for TPO positivity based on the 95th percentile of healthy control samples.

Anti-nuclear antigen (ANA) status was determined from plasma samples using a qualitative ELISA kit (MyBioSource, cat. no. 702970) according to manufacturer instructions, with a sample $OD_{450nm}$ / negative control $OD_{450nm}$ ratio ≥2.1 evaluated as positive and a ratio <2.1 evaluated as negative.

Autoantigen profiling of EDTA plasma samples (50 µL each; T21, n=120; D21, n=60) was performed by the Affinity Proteomics unit at SciLifeLab (KTH Royal Institute of Technology, Stockholm, Sweden) using peptide arrays. Antigens were selected to cover potential associations to autoimmune diseases and consisted of 380 peptide fragments covering ~270 unique proteins (1–5 fragments per protein). Fragments were ~20–163 amino acids long (median 82). All antigens were expressed in *E. coli* with a hexahistidyl and albumin binding protein tag (His6ABP). Using, COOH-NH2 chemistry, the analyzed antigens, in addition to controls, were immobilized on color-coded magnetic beads (MagPlex, Luminex). Controls consisted of His6ABP, buffer, rabbit anti-human IgG (loading control, Jackson ImmunoResearch), and Epstein-Barr nuclear antigen 1 (EBNA1, Abcam). Research samples and technical controls (commercial plasma; Seralab) were diluted (1:250) in assay buffer, which consisted of 3% BSA, 5% milk, 0.05% Tween-20, and 160 µg/ml His6ABP tag in PBS. Diluted samples and controls were incubated for 1 hr at room temperature then subsequently incubated with the antigen bead array for 2 hr. The reactions were then fixed for 10 min using 0.02% paraformaldehyde, then incubated for 30 min with goat Fab specific for human IgG Fc-γ tagged with the fluorescent marker R-phycoerythrin (Invitrogen). Median fluorescence intensity (MFI) and number of beads for each reaction was analyzed using a FlexMap 3D instrument (Luminex Corp.). Quality control was performed using MFI and bead count to exclude antigens and samples not passing technical criteria including minimal bead counts and antigen coupling efficiency. To adjust for sample specific backgrounds, MFI values were transformed per reaction median absolute deviations (MADs) using the following calculation:

$$MADs_{sample} = (MFI - median_{sample(MFI)})/MAD_{sample(MFI)}$$

Subsequent data analysis and handling was performed using R. For each antigen, positivity was defined as >90 th percentile MAD value for D21 samples only. Overrepresentation of positivity for each antigen in the T21 versus D21 group was determined using Fisher's exact test, excluding antigens detected in <18 samples (<10% of total experiment). Correction for multiple testing was performed using the Benjamini-Hochberg approach and significance defined as q<0.1 (10% FDR). Similarly, within the T21 group, Fisher's exact test was used to test for overrepresentation of antigen positivity in cases versus controls for co-occurring conditions, with only those with at least five cases considered in the analysis.

## Immune profiling via mass cytometry

Generation of the mass cytometry dataset was described previously (*Galbraith et al., 2023*), but a full description is included here for reference. Two 0.5 mL aliquots of EDTA whole blood samples underwent RBC lysis and white blood cell fixation using TFP FixPerm Buffer (Transcription Factor Phospho Buffer Set, BD Biosciences). WBCs were then washed in 1 x in PBS (Rockland), resuspended in Cell Staining Buffer (Fluidigm) and stored at –80 °C. For antibody staining, samples were thawed at room temperature, washed in Cell Staining Buffer, barcoded using a Cell-ID 20-Plex Pd Barcoding Kit (Fluidigm), and combined per batch. Each batch was able to accommodate 19 samples with a common

reference sample. Antibodies were either purchased pre-conjugated to metal isotopes or conjugation was performed in-house using a Maxpar Antibody Labeling Kit (Fluidigm). See Key Resources Table for antibodies. Working dilutions for antibody staining were titrated and validated using the common reference sample and comparison to relative frequencies obtained by independent flow cytometry analysis. Surface marker staining was carried out for 30 min at 4 °C in Cell Staining Buffer with added Fc Receptor Binding Inhibitor (eBioscience/ThermoFisher Scientific). Staining was followed by a wash in Cell Staining Buffer. Next, cells were permeabilized in Buffer III (Transcription Factor Phospho Buffer Set, BD Pharmingen) for 20 min at 4 °C followed by washing with perm/wash buffer (Transcription Factor Phospho Buffer Set, BD Pharmingen). Intracellular transcription factor and phospho-epitope staining was carried out for 1 hr at 4 °C in perm/wash buffer (Transcription Factor Phospho Buffer Set, BD Pharmingen), followed by a wash with Cell Staining Buffer. Cell-ID Intercalator-Ir (Fluidigm) was used to label barcoded and stained cells. Labeled cells were analyzed on a Helios instrument (Fluidigm). Mass cytometry data were exported as v3.0 FCS files for pre-processing and analysis.

## Analysis of mass cytometry data

### Pre-processing

Bead-based normalization via polystyrene beads embedded with lanthanides, both within and between batches, followed by bead removal was carried out as previously described using the Matlab-based Normalizer tool (*Finck et al., 2013*). Batched FCS files were demultiplexed using the Matlab-based Single Cell Debarcoder tool (*Zunder et al., 2015*). Reference-based normalization of individual samples across batches against the common reference sample was then carried out using the R script `BatchAdjust()`. For the analyses described in this manuscript, CellEngine (CellCarta) was used to gate and export per-sample FCS files at four levels: Firstly, CD3 +CD19+doublets were excluded and remaining cells exported as 'Live' cells; Live cells were then gated for hematopoietic lineage (CD45-positive) non-granulocytic (CD66-low) cells and exported as CD45 +CD66 low. Lastly, CD45 +CD66 low cells were gated on CD3-positivity and CD19-positivity and exported as T- and B-cells, respectively. Per-sample FCS files were then subsampled to a maximum of 50,000 events per file for subsequent analysis.

### Unsupervised clustering

For each of the four levels (live, non-granulocytes, T cells, and B cells), all 388 per-sample FCS files were imported into R as a flowSet object using the `read.flowSet()` function from the flowCore R package (*Hahne et al., 2009*). Next a SingleCellExperiment object was constructed from the flowSet object using the `prepData()` function from the CATALYST package (*Chevrier et al., 2018*). Arcsinh transformation was applied to marker expression data with cofactor values ranging from ~0.2 to~15 to give optimal separation of positive and negative populations for each marker, using the `estParam-FlowVS()` function from the flowVS R package (*Azad et al., 2016*) and based on visual inspection of marker histograms (see Key Resources Table). Quality control and diagnostic plots were examined with the help of functions from CATALYST and the tidySingleCellExperiment R package. Unsupervised clustering using the FlowSOM algorithm (*Van Gassen et al., 2015*) was carried out using the `cluster()` function from CATALYST, with grid size set to 10x10 to give 100 initial clusters and a maxK value of 40 was explored for subsequent meta-clustering using the ConsensusClusterPlus algorithm. Examination of delta area and minimal spanning tree plots indicated that 30–40 meta clusters gave a reasonable compromise between gains in cluster stability and number of clusters for each level. Each clustering level was re-run with multiple random seed values to ensure consistent results.

### Visualization using t-distributed stochastic neighbor imbedding (tSNE)

Dimensionality reduction to two dimensions was carried out using the `runDR()` function from the CATALYST package, with 500 cells per sample, and using several random seed values to ensure consistent results. Multiple values of the perplexity parameter were tested, with a setting of 440, using the formula Perplexity = $N^{(1/2)}$ as suggested at https://towardsdatascience.com/how-to-tune-hyper-parameters-of-tsne-7c0596a18868, providing a visualization with good agreement with the clusters defined by FlowSOM.

## Cell type classification

To aid in assignment of clusters to specific lineages and cell types, the MEM package (marker enrichment modeling) was used to call positive and negative markers for each cell cluster based on marker expression distributions across clusters. Manual review and comparison to marker expression histograms, as well as minimal spanning tree plots and tSNE plots colored by marker expression, allowed for high-confidence assignment of most clusters to specific cell types. Clusters that were insufficiently distinguishable were merged into their nearest cluster based on the minimal spanning tree. Relative frequencies for each cell type / cluster were calculated for each sample as a percentage of total live cells and as a percentage of cells used for each level of clustering: total CD45 +CD66 low cells, total T cells, or total B cells.

## Beta regression analysis

To identify cell clusters for which relative frequencies are associated with either trisomy 21 status or with various clinical subgroups (e.g. ANA+) among individuals with trisomy 21, beta regression analysis was carried out using the betareg R package, with each model using cell type cluster proportions (relative frequency) as the outcome/dependent variable and either T21 status or clinical subgroups as independent/predictor variables, along with adjustment for age and sex, and a logit link function. Extreme outliers were classified per-karyotype and per-cluster as measurements more than three times the interquartile range below or above the first and third quartiles, respectively (below Q1 - 3*IQR or above Q3 +3*IQR) and excluded from beta regression analysis. Correction for multiple comparisons was performed using the Benjamini-Hochberg (FDR) approach. Effect sizes (as fold-change in T21 vs. euploid controls or among T21 subgroups) for each cell type cluster were obtained by exponentiation of beta regression model coefficients. Fold-changes were visualized by overlaying on tSNE plots using ggplot2. For visualization of individual clusters, data points were adjusted for age and sex, using the `adjust()` function from the datawizard R package, and visualized as sina plots (separated by T21 status or clinical subgroup).

## Measurement of immune markers and calculation of cytokine scores

Briefly, from each EDTA plasma sample, two replicates of 12–25 µL were analyzed using the Meso Scale Discovery (MSD) multiplex immunoassay platform V-PLEX Human Biomarker 54-Plex Kit (HTP cohort) or U-PLEX Human Biomarker Group 1 71-Plex and V-PLEX Human Vascular Injury Panel 2 Kits (clinical trial cohort) on a MESO QuickPlex SQ 120 instrument. Assays were carried out as per manufacturer instructions. Concentration values were calculated against a standard curve with provided calibrators. MSD data are reported as concentration values in picograms per milliliter of plasma.

## Analysis of immune marker data

Plasma concentration values (pg/mL) for each of the cytokines and related immune factors measured across multiple MSD assay plates was imported to R, combined, and analytes with >10% of values outside of detection or fit curve range flagged. For each analyte, missing values were replaced with either the minimum (if below fit curve range) or maximum (if above fit curve range) calculated concentration per plate/batch and means of duplicate wells used for subsequent analysis. For the HTP study analysis, extreme outliers were classified per-karyotype and per-analyte as measurements more than three times the interquartile range below or above the first and third quartiles, respectively, and excluded from further analysis. Differential abundance analysis for inflammatory markers measured by MSD was performed using mixed effects linear regression as implemented in the `lmer()` function from the lmerTest R package (v3.1–2) with log2-transformed concentration as the outcome/dependent variable, T21 status or clinical subgroup (e.g., ANA+) as the predictor/independent variable, age and sex as fixed covariates, and sample source as a random effect. Multiple hypothesis correction was performed with the Benjamini-Hochberg method using a false discovery rate (FDR) threshold of 10% (q<0.1). Prior to visualization or correlation analysis, MSD data were adjusted for age, sex, and sample source using the `removeBatchEffect()` function from the limma package (v3.44.3).

### Calculation of cytokine scores

For comparison of clinical trial samples across time points, cytokine scores were calculated as the sum of the Z-scores for TNF-α, IL-6, CRP, and IP-10. For comparison of clinical trial samples to the HTP cohort, Z-scores were first calculated from age-, sex, and batch-adjusted values for each sample, based on the mean and standard deviation of the HTP euploid control samples.

## Whole blood transcriptome analysis and calculation of IFN scores

Strand-specific sequencing libraries were prepared from globin-depleted, polyA-enriched whole blood RNA and sequenced on the Illumina NovaSeq platform (2x150 bases). Data quality was assessed using FASTQC (v0.11.5) and FastQ Screen (v0.11.0). Trimming and filtering of low-quality reads was performed using bbduk from BBTools (v37.99) and fastq-mcf from ea-utils (v1.05). Alignment to the human reference genome (GRCh38) was carried out using HISAT2 (v2.1.0) in paired, spliced-alignment mode against a GRCh38 index and Gencode v33 basic annotation GTF, and alignments were sorted and filtered for mapping quality (MAPQ >10) using Samtools (v1.5). Gene-level count data were quantified using HTSeq-count (v0.6.1) with the following options (`--stranded=reverse` –minaqual = 10 –type = exon `--mode=intersection-nonempty`) using a Gencode v33 GTF annotation file. Differential gene expression in T21 versus D21 was evaluated using DESeq2 (version 1.28.1) *Love et al., 2014*, with q<0.1 (10% FDR) as the threshold for differential expression.

### DS IFN scores

RNA-seq-based 'Down syndrome interferon scores' (DS IFN scores) were calculated as follows: for comparison of clinical trial samples across time points, DS IFN scores were calculated as the sum of Z-scores across 16 interferon-stimulated genes (ISGs) genes with significant mean fold-change of at least 1.5 in the HTP T21 group vs. the euploid control group, excluding *IFNAR2*, *MX1*, and *MX2* which are encoded on chromosome 21. For comparison of clinical trial samples to the HTP cohort, gene-wise Z-scores were first calculated from age-, sex, and sequencing batch-adjusted FPKM values for each sample, based on the mean and standard deviation of the HTP euploid control samples.

*Gene set enrichment analysis (GSEA)*. GSEA (*Subramanian et al., 2005*) was carried out in R using the fgsea package (v1.14.0), using Hallmark gene sets, $\log_2$-transformed fold-change values as the ranking metric.

## Clinical trial design and oversight

All aspects of this study were conducted in accordance with the Declaration of Helsinki. All study activities were approved by the Colorado Multiple Institutional Review Board (COMIRB, protocol # 19–1362, NCT04246372) with an independent Data and Safety Monitoring Board (DSMB) appointed by the National Institute of Arthritis and Musculoskeletal and Skin Diseases (NIAMS). Written consent was obtained from all participants, or their legally authorized representative if the participant was unable to provide consent, in which case participant assent was obtained. The Clinical Trial Protocol is provided in *Reporting standard 1*. We report here interim results of a single-site, open-label phase 2 clinical trial enrolling individuals with DS between the ages of 12 and 50 years old with moderate-to-severe alopecia areata, hidradenitis suppurativa, psoriasis, atopic dermatitis, or vitiligo. After screening, qualifying participants are prescribed 5 mg tofacitinib twice daily for 16 weeks, with an optional extension arm to week 40. During the main 16-week trial, participants attend five safety monitoring visits after enrollment at the Baseline visit.

## Trial population

The recruitment goal for this trial is 40 participants completing 16 weeks of tofacitinib treatment, with a qualitative interim analysis triggered when 10 participants completed the main 16 week trial. Of the 10 participants included in the interim analysis, 4 are female, 100% identify as White/Caucasian, 3 identify as Hispanic or Latino, and mean age at enrollment was 23.1 years old (range 15–38.1 years old). Baseline qualifying conditions of the 10 participants were alopecia areata: n=6 (46.1%), hidradenitis suppurativa: n=3 (30.8%), or psoriasis: n=1 (7.7%). Two participants also had atopic dermatitis, and two others had vitiligo, albeit below the severity required to be the qualifying condition.

## Outcome measures

### Primary endpoints

The two primary outcome measures for this trial are safety and reduction in IFN transcriptional scores derived from peripheral whole blood. Based on the safety profile for tofacitinib in the general population (*Ytterberg et al., 2022*), the safety primary endpoint was defined as no more than two serious adverse events (SAEs) definitely attributable to tofacitinib over the course of 16 weeks for 40 participants. Adverse events were classified based using Common Terminology Criteria for Adverse Events 5.0 (CTCAE 5.0). IFN scores are commonly used to monitor disease severity and response to treatment in IFN-driven pathologies (*Banchereau et al., 2016*; *de Jesus et al., 2020*) and their calculation form RNAseq data is described above.

### Secondary endpoints

The secondary outcome measures for this trial include improvements in skin health as defined by a global assessment, the Investigator Global Assessment (IGA), as well as the disease-specific assessments. Overall skin pathology, accounting for all present skin conditions regardless of severity, was assessed using a modified IGA which scores on a five-point scale for each skin condition (six points for HS) with a range of 0–21. Another secondary endpoint assessing global skin health is a change in the Dermatological Quality of Life Index (DLQI), used to assess participant-reported impact of skin conditions on self-image, relationships, and daily activities. Possible total scores range from 0 to 30, with higher scores indicating a more impaired quality of life. Condition-specific assessments used are Severity of Alopecia Tool (SALT) for AA affecting at least 25% of the scalp (qualifying score is ≥25); Hidradenitis Suppurativa-Physicians Global Assessment (HS-PGA) to define eligibility (qualifying score ≥3) and Modified Sartorius Scale (MSS) to monitor changes throughout the study for HS; Psoriasis Area and Severity Index (PASI, qualifying score is ≥10) for psoriasis; Vitiligo Extent Tensity Index (VETI, qualifying score is ≥2), for moderate-to-severe vitiligo; and Eczema Area and Severity Index (EASI, qualifying EASI score ≥16) for moderate-to-severe atopic dermatitis. The last secondary endpoint is reduction in a cytokine score coalescing information on four inflammatory markers elevated in DS: Tumor Necrosis Factor Alpha (TNF-α), interleukin 6 (IL-6), C-reactive protein (CRP), and IFN-inducible protein 10 (IP10, CXCL10; *Sullivan et al., 2017*). Measurement of these proteins and calculation of the cytokine score is described above.

### Tertiary endpoints

This clinical trial includes multiple exploratory tertiary endpoints (see full protocol in *Reporting standard 1*), including reduction in autoantibodies related to AITD (anti-TPO, anti-TG, and anti-TSHr) and celiac disease (anti-tTG, anti-DGP). In the clinical trial, these autoantibodies were assessed using established clinical assays.

## Statistical analysis

The Statistical Analysis Plan (SAP) approved by the appointed DSMB is included with the Clinical Trial Protocol in *Reporting standard 1*. This report includes analysis of the time points used to assess endpoints (baseline and 16 weeks), as well research-only time points at 2 and 8 weeks of treatment. Given the qualitative nature of this interim analysis, statistical analysis is not completed for changes observed between baseline and the 16-week endpoint. Data may be displayed as $log_2$ transformed for clarity in viewing the graphs.

### Code Availability Statement

No custom code or algorithms were developed during the course of this study. Software packages are listed in the Key Resources Table. R analysis scripts will be made available upon request.

## Results

### Widespread multi-organ autoimmunity and autoantibody production in Down syndrome

Previous studies have documented increased rates of diverse autoimmune conditions in DS relative to the general population including autoimmune thyroid disease (AITD; *Aversa et al., 2018*), celiac disease (*Liu et al., 2020*), autoimmune skin conditions (*Madan et al., 2006*; *Sureshbabu et al., 2011*; *Lam et al., 2020*), and type I diabetes (*Aitken et al., 2013*; *Johnson et al., 2019*). However, many of these studies were limited by relatively small sample sizes, independent analysis of individual autoimmune conditions, or a focus on specific age ranges. In order to complete a more comprehensive analysis of autoimmune conditions in DS across the lifespan, we analyzed the harmonized clinical profiles of 441 research participants with DS, aged 6 months to 57 years, enrolled in the Human Trisome Project cohort study (HTP, NCT02864108), which annotates clinical data through a combination of participant/caregiver surveys and expert abstraction of electronic health records (EHRs; see Materials and methods, *Supplementary file 1*). In this analysis, the most common autoimmune condition is AITD, affecting 53.1% of the total cohort (*Figure 1a*, *Figure 1—figure supplement 1a*, *Figure 1—source data 1*). Grouped together, autoimmune and inflammatory skin conditions represent the second most common category, affecting 43% of the cohort, including: atopic dermatitis / eczema (27.9%), hidradenitis suppurativa / folliculitis / boils (20.6%), alopecia areata (7.7%), psoriasis (6.1%), and vitiligo (1.9%; *Figure 1a*, *Figure 1—figure supplement 1b*, *Figure 1—source data 1*). These observations align with recent epidemiological studies demonstrating high rates of autoimmune and inflammatory skin conditions in DS (*Gansa et al., 2024*; *Rakasiwi et al., 2024*). The rate of celiac disease (9.6%) is also highly elevated over that of the general population (*Singh et al., 2018*). We observed 10 cases (2.2%) of juvenile Type I diabetes, which has been reported to be more common in DS (*Aitken et al., 2013*; *Johnson et al., 2019*). Other autoimmune conditions common in the general population, such as systemic lupus erythematosus or multiple sclerosis, were not observed in the HTP cohort. Other salient conditions annotated in this cohort include recurrent otitis media (15.5%), frequent/recurrent pneumonia (9.2%), severe congenital heart defects requiring surgical repair (19.5%), acute lymphocytic leukemia (ALL, 1.12%), and acute myeloid leukemia (AML, 1.3%; *Figure 1—source data 1*).

In the general population, the risk of autoimmune conditions increases with age and is higher in females, with autoimmune conditions tending to cluster, whereby occurrence of one autoimmune condition predisposes to a second condition (*Markle et al., 2013*; *Molano-González et al., 2019*). Within the HTP cohort, analysis of age trajectories of immune-related conditions in DS revealed early onset, with >80% of AITD, autoimmune/inflammatory skin conditions, and celiac disease being diagnosed in the first two decades of life (*Figure 1—figure supplement 1c–e*). The cumulative burden of autoimmunity and autoinflammation is similar in males versus females with DS, albeit with slightly increased rates of AITD and hidradenitis suppurativa in females (*Figure 1—figure supplement 1f*, *Figure 1—source data 1*). In terms of co-occurrence, when evaluating the adult population (18+years old) for AITD, autoimmune/inflammatory skin conditions and celiac disease, we found that 75% of participants had a history of at least one condition, 38.4% had at least two, and 13.6% had three or more conditions (*Figure 1b*, *Figure 1—source data 1*).

Interestingly, analysis of medical records found an unexpectedly low number of individuals with records of autoantibodies against the thyroid gland (4.3%, e.g., anti-thyroid peroxidase [TPO], anti-thyroglobulin [TG]) within the HTP cohort (*Figure 1—figure supplement 1a*). This could be explained by the fact that thyroid disease is commonly diagnosed through measurements of thyroid-relevant hormones (TSH, T3, T4) without concurrent testing of autoantibodies. To investigate further, we measured anti-TPO levels as well as levels of anti-nuclear antibodies (ANA), a more general biomarker of autoimmunity. Remarkably, 82.4% of adults with DS show positivity for at least one of these autoantibodies, with 41.2% being positive for both (*Figure 1c*, *Figure 1—source data 2*). Indeed, 62% of individuals with history of hypothyroidism were TPO+, whereby anti-TPO is just one of the possible autoantibodies associated with AITD. Prompted by these results, we next completed a more comprehensive analysis of autoantibodies in DS using protein array technology, with a focus on ~380 common autoepitopes from 270 proteins (see Materials and methods, *Figure 1—source data 2*). These efforts identified 25 autoantibodies significantly over-represented in people with DS relative to age- and sex-matched controls (*Figure 1d*), with 98.3% of individuals with DS being positive for at least one of these autoantibodies, and 63.3% being positive for six or more (*Figure 1d–e*). In addition

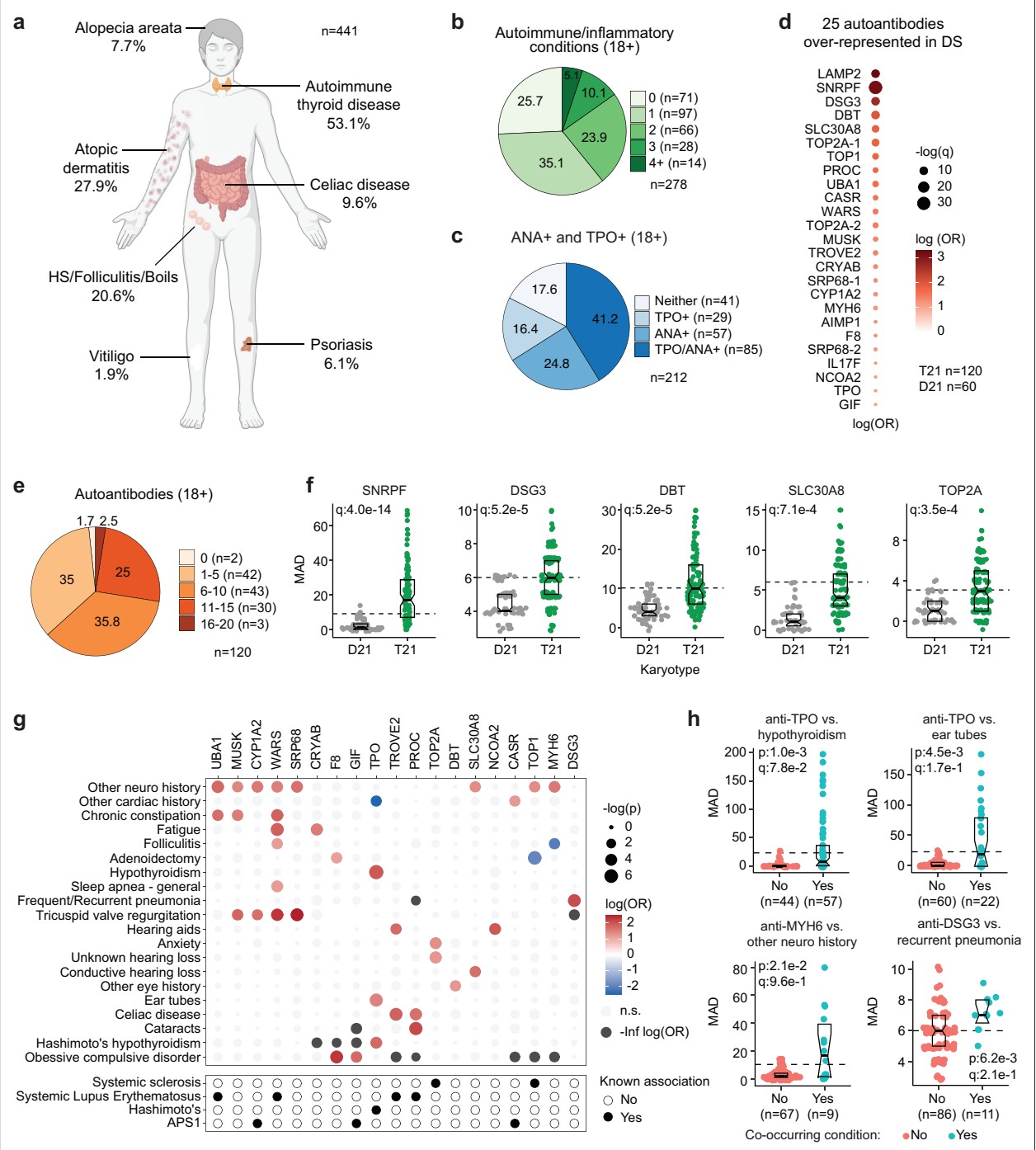

**Figure 1.** Multi-organ autoimmunity and widespread autoantibody production in Down syndrome. (**a**) Overview of autoimmune and inflammatory conditions prevalent in persons with Down syndrome (DS) enrolled in the Human Trisome Project (HTP) cohort study. Percentages indicate the fraction of participants (n=441, all ages) with history of the indicated conditions. Graphic elements composed with BioRender.com. (**b**) Pie chart showing autoimmune/inflammatory condition burden in adults (n=278, 18+years old) with DS. (**c**) Pie chart showing rates of positivity for anti-TPO and/or anti-nuclear antibodies (ANA) in adults (n=212, 18+years old) with DS. (**d**) Bubble plot displaying odds-ratios and significance for 25 autoantibodies with elevated rates of positivity in individuals with DS (n=120) vs 60 euploid controls (D21). q values calculated by Benjamini-Hochberg adjustment of p-values from Fisher's exact test. (**e**) Pie chart showing fractions of adults with DS (n=120, 18+years old) testing positive for various numbers of the autoantibodies identified in d. (**f**) Representative examples of autoantibodies more frequent in individuals with T21 (n=120) versus euploid controls (D21, n=60). MAD: median absolute deviation. Dashed lines indicate the positivity threshold of 90th percentile for D21. Data are presented as modified sina plots with boxes indicating quartiles. (**g**) Bubble plots showing the relationship between autoantibody positivity and history of various clinical diagnoses in DS (n=120). Size of bubbles is proportional to -log-transformed p values from Fisher's exact test. (**h**) Sina plots displaying the levels of selected

*Figure 1 continued on next page*

*Figure 1 continued*

autoantibodies in individuals with DS with or without the indicated co-occurring conditions. MAD: median absolute deviation. Dashed lines indicate the positivity threshold of 90th percentile for D21. Sample sizes are indicated under each plot. q values calculated by Benjamini-Hochberg adjustment of p-values from Fisher's exact tests.

The online version of this article includes the following source data and figure supplement(s) for figure 1:

**Source data 1.** Clinical data for Human Trisome Project participants analyzed in this study, including demographics, karyotype status, and major co-occurring diagnoses relevant to this study.

**Source data 2.** Autoantibody measurements of Human Trisome Project participants.

**Figure supplement 1.** Early onset multi-organ autoimmunity and autoantibody production in Down syndrome.

---

to autoantibodies against TPO, which is expressed exclusively in the thyroid gland, we identified autoantibodies targeting proteins that are either broadly expressed across multiple tissues (e.g. TOP1, UBA1, LAMP2) or preferentially expressed in specific organs across the human body, including liver (e.g. CYP1A2), pancreas (e.g. SLC30A8), skin (e.g. DSG3), bone marrow (e.g. SRP68), and brain tissue (e.g. AIMP1; *Figure 1d and f*).

Analysis of autoantibody positivity relative to history of co-occurring conditions produced several interesting observations. Expectedly, individuals with hypothyroidism are more likely to be positive for anti-TPO antibodies (*Figure 1g–h*). However, unexpectedly, TPO+ status also associates with higher rates of use of pressure equalizing (PE) tubes employed to alleviate the symptoms of recurrent ear infections and otitis media with effusion (OME), which is common in DS (*Elling et al., 2023*, *Figure 1g–h*). Possible interpretations for this result are provided in the Discussion. Positivity for additional autoantibodies was more common in those with other co-occurring neurological conditions, a broad classification encompassing various seizure disorders, movement disorders, and structural brain abnormalities (*Figure 1g–h*, *Figure 1—figure supplement 1g*). Salient examples are antibodies against MUSK, a muscle-associated receptor tyrosine kinase involved in clustering of the acetylcholine receptors in the neuromuscular junction (*Ghazanfari et al., 2014*); UBA1, a ubiquitin conjugating enzyme involved in antigen presentation (*Poulter et al., 2021*); and MYH6, a cardiac myosin heavy chain isoform (*Figure 1g-h*, *Figure 1—figure supplement 1g*). Individuals with history of tricuspid valve regurgitation display higher rates of four different autoantibodies, most prominently against WARS1, a tryptophan tRNA synthetase mutated in various neurodevelopmental disorders (*Lin et al., 2022*), and SRP68, a protein commonly targeted by autoantibodies in necrotizing myopathies (*Allenbach et al., 2020*, *Figure 1—figure supplement 1g*). Individuals with a history of frequent pneumonia present a higher frequency of autoantibodies against DSG3 (desmoglein 3), a cell adhesion molecule targeted by autoantibodies in paraneoplastic pemphigus (PNP), an autoimmune disease of the skin and mucous membranes that can involve fatal lung complications (*Amagai et al., 1998*, *Figure 1g–h*).

Altogether, these results demonstrate widespread multi-organ autoimmunity across the lifespan in people with DS, with production of multiple autoantibodies that could potentially contribute to a number of co-occurring conditions more common in this population.

## Trisomy 21 causes global immune remodeling regardless of evident clinical autoimmunity

Several immune cell changes have been proposed to underlie the autoimmunity-prone state of DS (*Waugh et al., 2019*; *Araya et al., 2019*; *Lambert et al., 2022*; *Malle et al., 2023*), but specific immune cell-to-phenotype associations have not been established in previous studies using smaller sample sizes. Therefore, we next investigated immune cell changes associated with various clinical and molecular markers of autoimmunity in DS. Toward this end we analyzed mass cytometry data from 292 individuals with DS relative to 96 euploid controls and tested for potential differences in immune cell subpopulations, identified using FlowSOM (*Van Gassen et al., 2015*), within the DS cohort based on number of autoimmune/inflammatory disease diagnoses, ANA positivity, TPO positivity, and positivity for additional autoantibodies. In agreement with previous analyses (*Waugh et al., 2019*; *Araya et al., 2019*; *Lambert et al., 2022*; *Galbraith et al., 2023*; *Malle et al., 2023*), we observed massive immune remodeling in all major myeloid and lymphoid subsets, including increases in basophils, along with depletion of eosinophils and total B cells (*Figure 2a-c*, *Figure 2—figure supplement 1a-c*). When comparing various subgroups within the DS cohort based on autoimmunity

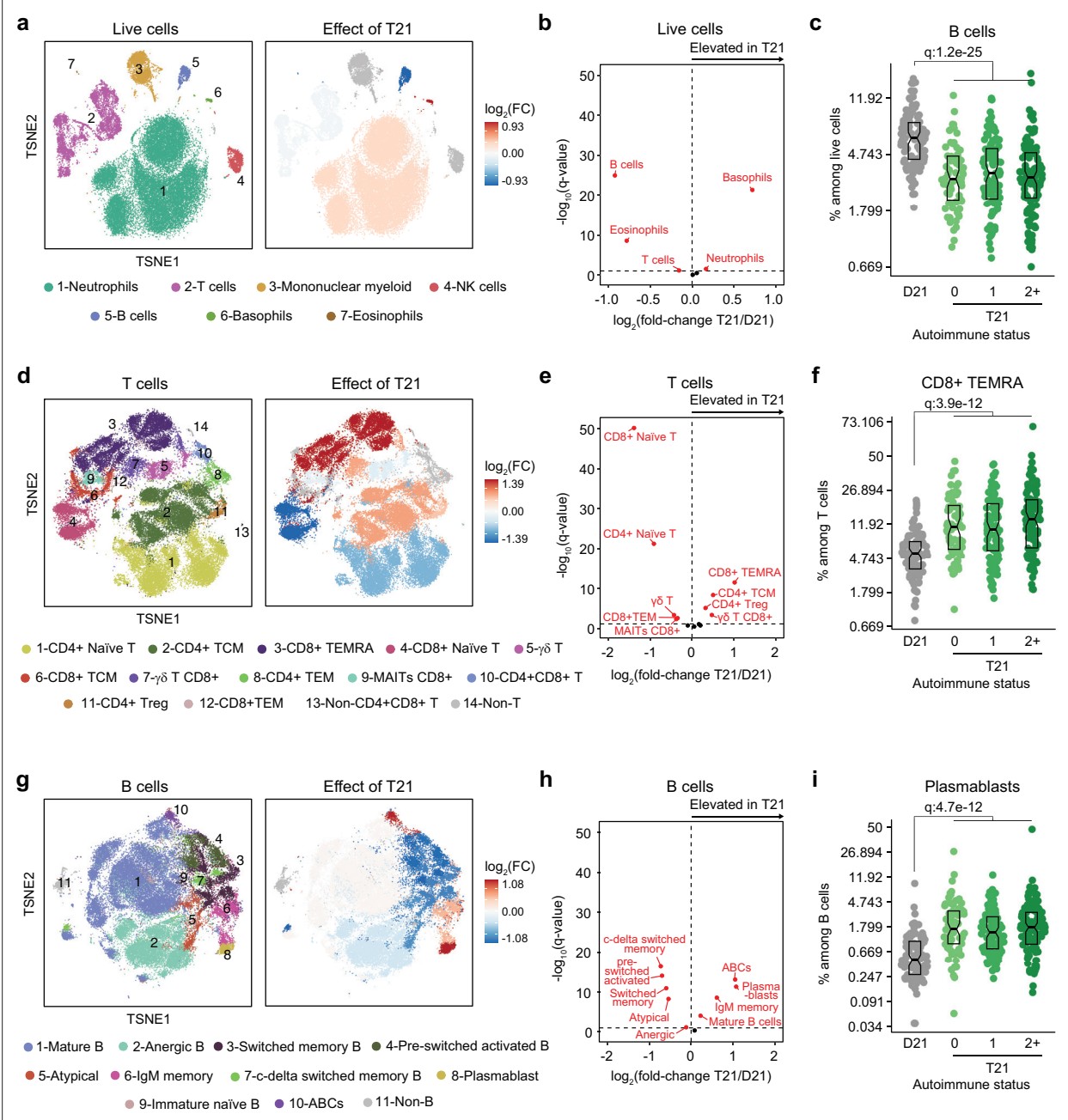

**Figure 2.** Trisomy 21 causes global immune remodeling regardless of clinically evident autoimmunity. (**a**) t-distributed Stochastic Neighbor Embedding (t-SNE) plot displaying major immune populations identified by FlowSOM analysis of mass cytometry data for all live cells (left) and color coded by significant impact of T21 (beta regression q<0.1) on their relative frequency (right). Red indicates increased frequency and blue indicates decreased frequency among research participants with T21 (n=292) versus euploid controls (D21, n=96). (**b**) Volcano plot showing the results of beta regression analysis of major immune cell populations among all live cells in research participants with T21 (n=292) versus euploid controls (D21, n=96). The dashed horizontal line indicates a significance threshold of 10% FDR (q<0.1) after Benjamini-Hochberg correction for multiple testing. (**c**) Frequencies of B cells among all live cells in euploid controls (D21, n=96) versus individuals with T21 and history of 0 (n=69), 1 (n=102) or 2+ (n=121) autoimmune/inflammatory conditions. Data is displayed as modified sina plots with boxes indicating quartiles. (**d-f**) Description as in a-c, but for subsets of T cells. (**g–i**) Description as in a-c, but for subsets of B cells.

The online version of this article includes the following figure supplement(s) for figure 2:

**Figure supplement 1.** Consistent remodeling of the peripheral immune system in Down syndrome.

status, we observed that these global immune changes are largely independent of the presence of clinical diagnoses or autoantibody positivity, with very few additional changes significantly associated with these measures of autoimmunity (*Figure 2—figure supplement 1d*). For example, the significant depletion of B cells and enrichment of basophils in DS is not significantly different among the various subgroups (*Figure 2c*, *Figure 2—figure supplement 1c*). Among CD45+ CD66^lo non-granulocytes, most changes are conserved among subgroups, with the sole of exception of non-classical mono-cytes, which are further elevated in the ANA+ group (*Figure 2—figure supplement 1d–e*). Among T cells, the overall pattern of depletion of naïve subsets and enrichment of differentiated subsets char-acteristic of DS (*Waugh et al., 2019*; *Araya et al., 2019*; *Malle et al., 2023*; *Galbraith et al., 2023*) is conserved across subgroups, as illustrated by consistent depletion of CD8+ naive subsets along with increases in the CD8+ terminally differentiated effector memory (TEMRA) subset (*Figure 2f*, *Figure 2—figure supplement 1d*). Notably, we observed depletion of γδ T cells (both total and CD8+) in those with multiple autoimmune diagnoses (*Figure 2—figure supplement 1d,f*), a result that is in line with reports documenting depletion of these subsets from peripheral circulation toward sites of active autoimmunity (*Amini et al., 2020*). We also observed slight elevation of CD4+ T central memory cells (TCM; *Figure 2—figure supplement 1d,f*). Among B cells, the overall shift toward more differentiated states such as plasmablasts, age-associated B cells (ABCs), and IgM+ memory cells is also conserved among subgroups, with the sole exception of ABCs, which tend to be further elevated in the TPO+ group (*Figure 2g–i*, *Figure 2—figure supplement 1d,g*).

Altogether, these results indicate that T21 causes global remodeling of the immune system toward an autoimmunity-prone and pro-inflammatory state, prior to clinically evident autoimmunity, and dwarfing any additional effects associated with confirmed diagnoses of autoimmune/inflammatory conditions or common biomarkers of autoimmunity.

## Trisomy 21 causes hypercytokinemia from an early age independent of autoimmunity status

It is well established that individuals with DS display elevated levels of many inflammatory markers, including several interleukins, cytokines, and chemokines known to drive autoimmune conditions, such as IL-6 and TNF-α (*Sullivan et al., 2017*; *Zhang et al., 2017*; *Malle et al., 2023*; *Galbraith et al., 2023*). However, the interplay between hypercytokinemia, individual elevated cytokines, and development of autoimmune conditions in DS remains to be elucidated. Therefore, we analyzed data available from the HTP cohort for 54 inflammatory markers in plasma samples from 346 individuals with DS versus 131 euploid controls and cross-referenced these data with the presence of autoim-mune conditions and autoantibodies. These efforts confirmed the notion of profound hypercytokin-emia in DS (*Sullivan et al., 2017*; *Zhang et al., 2017*; *Malle et al., 2023*; *Galbraith et al., 2023*), with significant elevation of multiple acute phase proteins (e.g. CRP, SAA, IL1RA), pro-inflammatory cyto-kines (TSLP, IL-17C, IL-22, IL-17D, IL-9, IL-6, TNF-α) and chemokines (IP-10, MIP-3a, MIP-1a, MCP-1, MCP-4, Eotaxin), as well as growth factors associated with inflammation and wound healing (FGF, PIGF, VEGF-A; *Figure 3a*). However, when evaluating for differences within the DS cohort based on various metrics of autoimmunity, we did not observe important differences based on number of autoimmune/inflammatory conditions, ANA or TPO positivity status, or number of other autoanti-bodies (*Figure 3a*). For example, CRP, IL-6, and TNF-α are equally elevated across all these subgroups (*Figure 3b–d*, *Figure 3—figure supplement 1a*).

Previous studies have reported signs of early immunosenescence and inflammaging in DS, including accelerated progression of immune lineages toward terminally differentiated states, early thymic atrophy, and elevated levels of pro-inflammatory markers associated with age in the typical population (*Kusters et al., 2010*; *Trotta et al., 2011*; *Gensous et al., 2020*; *Lambert et al., 2022*). However, the extent to which the inflammatory profile of DS represents accelerated ageing versus other processes remains ill-defined. To address this, we first identified age-associated changes in immune markers within the euploid and DS cohorts separately (*Figure 3—figure supplement 1b–c*). This exercise identified multiple immune markers that were up- or down-regulated with age, with an overall conserved pattern of age trajectories in both groups (*Figure 3—figure supplement 1c*). For example, increased age is associated with increased CRP levels and decreased IL-17B levels in both cohorts (*Figure 3—figure supplement 1d*). We then compared the effects of age versus T21 status on cytokine levels in the DS cohort, which identified many inflammatory factors elevated in DS across

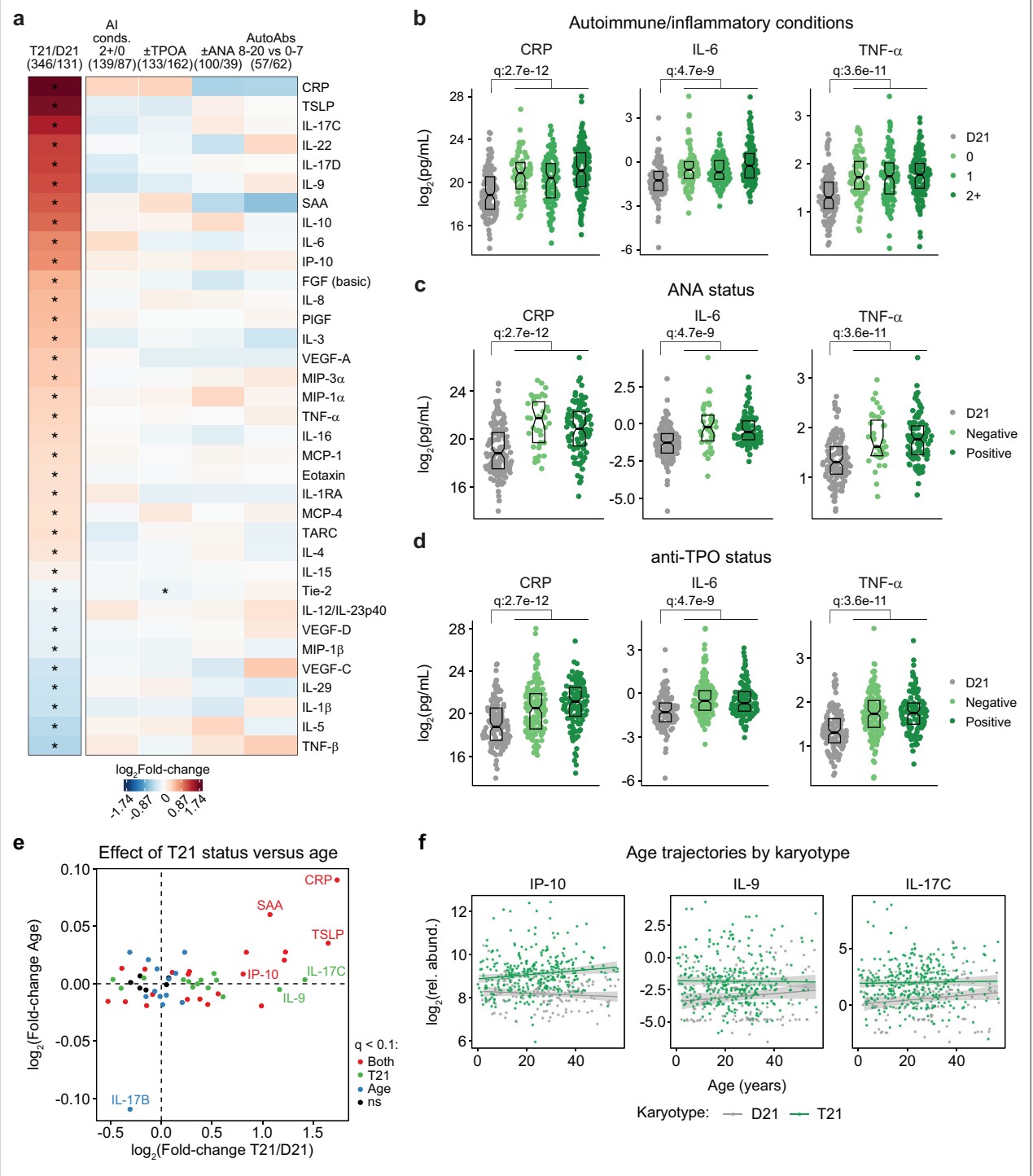

**Figure 3.** Trisomy 21 causes constitutive hypercytokinemia independent of autoimmunity status from an early age. (**a**) Heatmap displaying log₂-transformed fold-changes for plasma immune markers with significant differences in trisomy 21 (T21, n=346) versus euploid (D21, n=131), and between different subgroups within the T21 cohort: history of 2+ (n=139) vs 0 (n=87) autoimmune/inflammatory conditions (AI conds.); TPO+ (n=133) versus TPO- (n=162); ANA+ (n=100) versus ANA- (n=39); or positivity for 8–20 (n=57) versus 0–7 (n=62) autoantibodies (AutoAbs) elevated in DS. Asterisks indicate linear regression significance after Benjamini-Hochberg correction for multiple testing (q<0.1, 10% FDR). (**b–d**) Comparison of CRP, IL-6 and TNF-α levels in euploid controls (D21, n=131) versus subsets of individuals with T21 based on number of autoimmune/inflammatory conditions (**b**), ANA positivity (**c**) or TPO positivity (**d**). Data are presented as modified sina plots with boxes indicating quartiles. Samples sizes as in a. q-values indicate linear regression significance after Benjamini-Hochberg correction for multiple testing. (**e**) Scatter plot comparing the effect of T21 karyotype versus the effect of age in individuals with T21 (n=54 immune markers in 346 individuals with T21), highlighting immune markers that are significantly different by

*Figure 3 continued on next page*

*Figure 3 continued*

T21 status, age, or both. ns: not significantly different by T21 status or age. (**f**) Scatter plots for example immune markers that are significantly elevated in T21, but which are either not elevated with age in the euploid (D21) cohort (i.e. IP-10), or in either the T21 (n=346) or D21 (n=131) cohorts. Lines represent least-squares linear fits with 95% confidence intervals in grey.

The online version of this article includes the following figure supplement(s) for figure 3:

**Figure supplement 1.** Consistent hypercytokinemia from an early age in Down syndrome.

the lifespan that do not display a significant increase with age, such as IL-9 and IL-17C, or that increase with age only in the DS cohort, such as IP-10 (*Figure 3e–f*, *Figure 3—figure supplement 1e*).

Altogether, these results indicate that T21 induces a constitutive hypercytokinemia from early childhood, with only a fraction of these inflammatory changes being exacerbated with age.

## A clinical trial for JAK inhibition in Down syndrome

Several lines of evidence support the notion that IFN hyperactivity and downstream JAK/STAT signaling are key drivers of immune dysregulation in DS (*Sullivan et al., 2016*; *Sullivan et al., 2017*; *Waugh et al., 2019*; *Araya et al., 2019*; *Powers et al., 2019*; *Tuttle et al., 2020*; *Chi et al., 2023*; *Galbraith et al., 2023*; *Waugh et al., 2023*). In mouse models of DS, both normalization of *IFNR* gene copy number and pharmacologic JAK1 inhibition rescue their lethal immune hypersensitivity phenotypes (*Tuttle et al., 2020*; *Waugh et al., 2023*). Furthermore, we recently demonstrated that IFN transcriptional scores derived from peripheral immune cells correlate significantly with the degree of immune remodeling and hypercytokinemia in DS (*Galbraith et al., 2023*), and we and others have reported the safe use of JAK inhibitors for treatment of diverse immune conditions in DS, including alopecia areata (*Rachubinski et al., 2019*), psoriatic arthritis (*Pham et al., 2021*) and hemophagocytic lymphohistiocytosis (*Guild et al., 2022*) through small case series. Encouraged by these results, we launched a clinical trial to assess the safety and efficacy of the JAK inhibitor tofacitinib (Xeljanz, Pfizer) in DS, using moderate-to-severe autoimmune/inflammatory skin conditions as a qualifying criterion (NCT04246372). This trial is a single-site, open-label, Phase II clinical trial enrolling individuals with DS between the ages of 12 and 50 years old affected by alopecia areata, hidradenitis suppurativa, psoriasis, atopic dermatitis, or vitiligo (see qualifying disease scores in *Supplementary file 2*). After screening, qualifying participants are prescribed 5 mg of tofacitinib twice daily for 16 weeks, with an optional extension to 40 weeks (*Figure 4a*, see Materials and methods, see protocol in *Reporting standard 1*). After enrollment and assessments at a baseline visit, participants attend five safety monitoring visits during the main 16-week trial period. The recruitment goal for this trial is 40 participants who complete 16 weeks of tofacitinib treatment, with a predefined IRB-approved qualitative interim analysis triggered when the first 10 participants completed the main 16-week trial (*Figure 4b*). Among the first 13 participants enrolled, one participant withdrew shortly after enrollment, one was excluded from analyses due to medication non-compliance (i.e. >15% missed doses), and one participant had not yet completed the trial at the time of the interim analysis (*Figure 4b*). Demographic characteristics of the 10 participants included in the interim analysis are shared in *Supplementary file 2*. Baseline qualifying conditions of the 10 participants included in the interim analysis were alopecia areata (n=6), hidradenitis suppurativa (n=3), and psoriasis (n=1; open circles in *Figure 4c*). Two participants presented with concurrent atopic dermatitis, two with concurrent vitiligo, and two with concurrent hidradenitis suppurativa, albeit below the severity required to be the qualifying conditions (see closed circles in *Figure 4c*). In addition, seven participants had AITD/TPO+ and three had a celiac disease diagnosis (*Figure 4c*).

## Tofacitinib is well tolerated in Down syndrome

Analysis of adverse events (AEs) recorded for the 10 first participants over 16 weeks did not identify any AEs considered definitely related to tofacitinib treatment or classified as severe. Several AEs were annotated as 'possibly related' to treatment (*Figure 4d*, *Figure 4—source data 1*). Five episodes of upper respiratory infections (URIs) affecting five different participants were observed. Based on the safety data for tofacitinib in the general population (*Cohen et al., 2017*), all episodes of URIs were annotated as possibly related to treatment. Participant AA2 developed occasional cough and rhinorrhea that resolved with over-the-counter medication. Two other participants reported transient

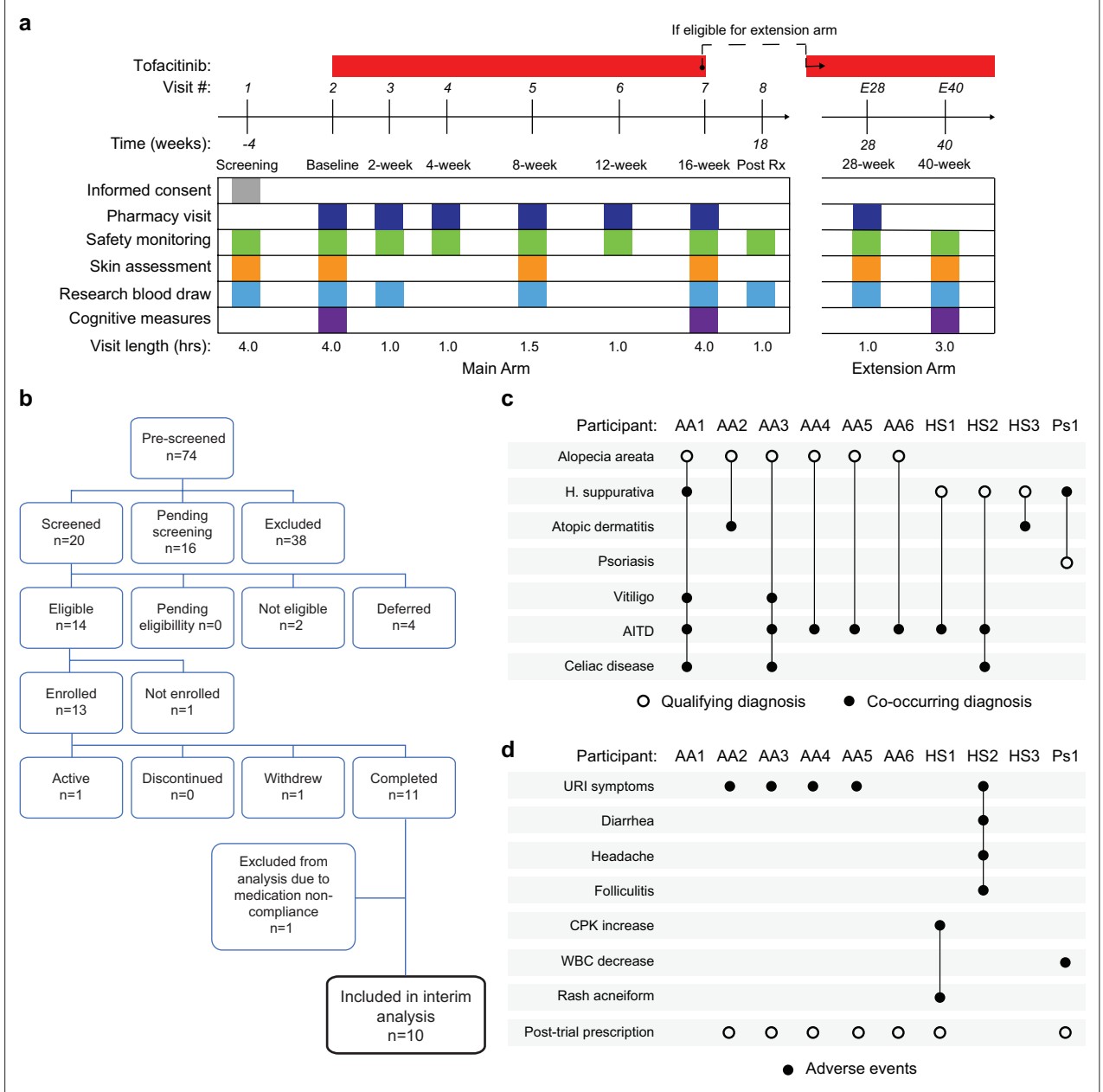

**Figure 4.** Clinical trial for JAK inhibition in Down syndrome. (**a**) Schedule of activities for clinical trial of JAK inhibition in Down syndrome (NCT04246372). (**b**) Consort chart for first 13 participants enrolled in the clinical trial. (**c**) Upset plot displaying the qualifying and co-occurring autoimmune/inflammatory conditions for the 10 participants included in the interim analysis. (**d**) Upset plots summarizing the adverse events annotated for the first 10 participants over a 16-week treatment period.

The online version of this article includes the following source data for figure 4:

**Source data 1.** Adverse events for clinical trial participants.

rhinorrhea (AA3, AA5). Participant AA4 developed a nasal congestion, with chest pain and a productive cough. This participant tested negative for SARS-Co-V2, Flu A-B, and RSV. Tofacitinib was not paused during this episode, and symptoms resolved with over-the-counter medication. Participant HS2 experienced a sore throat with middle ear inflammation that resolved with over-the-counter treatment. This participant also presented with folliculitis, which resolved with antibiotic treatment. Participant HS1 experienced a short transient elevation (<3 days) in creatine phosphokinase (CPK) that resolved spontaneously, and rash acneiform. Participant Ps1 experienced a transient and asymptomatic decrease in white blood cell (WBC) counts that resolved by the end of the trial.

Overall, tofacitinib treatment was not discontinued for any of the 10 participants over the 16-week study period, and seven participants eventually obtained off-label prescriptions after completing the trial and are currently taking the medicine. Based on these interim results, recruitment resumed and is ongoing.

## Tofacitinib improves diverse autoimmune/inflammatory skin conditions in Down syndrome

In the clinical trial, skin pathology is monitored using global metrics of skin health, including the Investigator's Global Assessment (IGA) and the Dermatology Life Quality Index (DLQI), as well as disease-specific scores, such as the severity of alopecia tool (SALT), the psoriasis area and severity

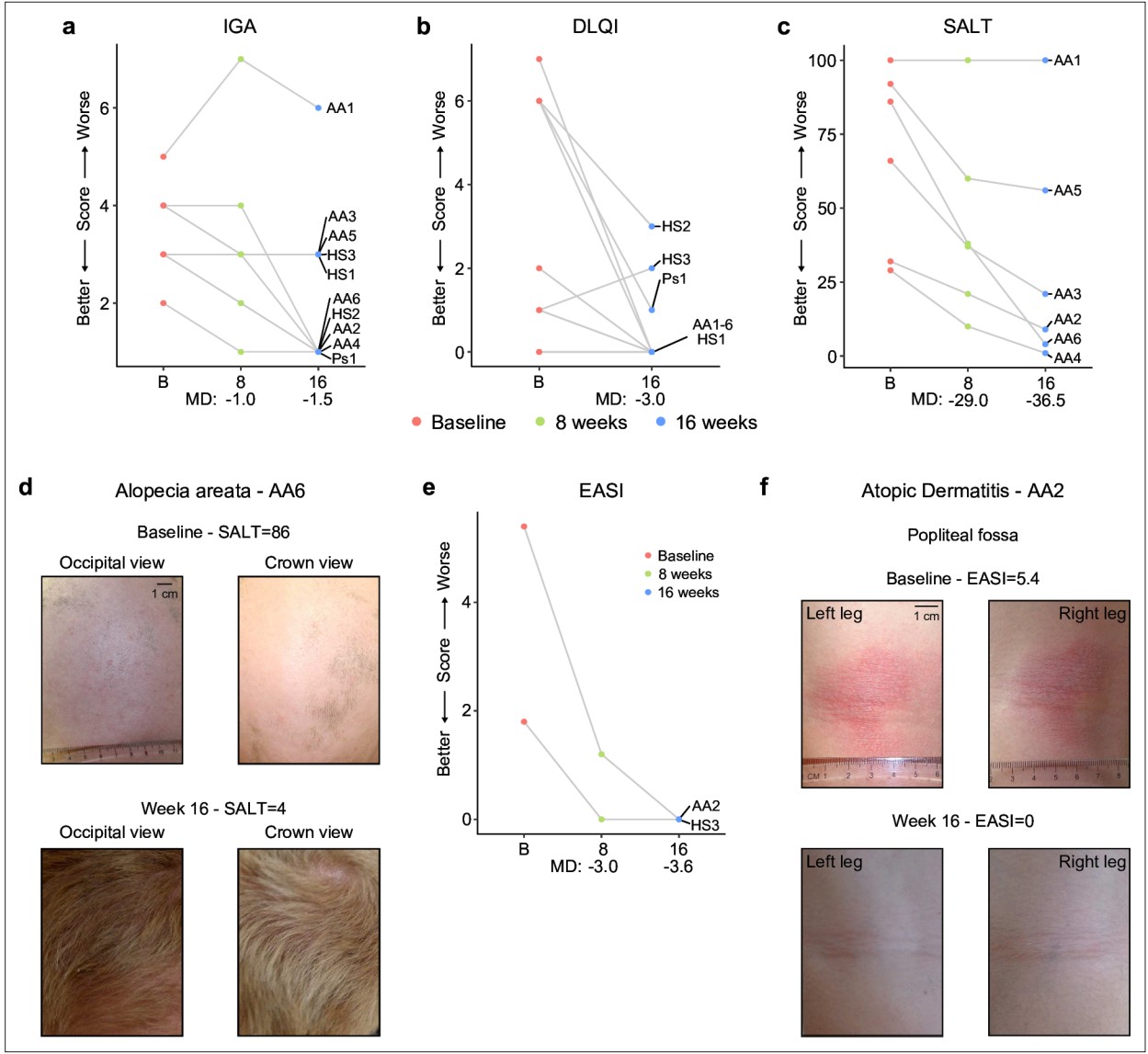

**Figure 5.** Tofacitinib improves diverse immune skin pathologies in Down syndrome. (**a–b**) Investigator global assessment (IGA) scores (**a**) and Dermatological Life Quality Index (DLQI) scores (**b**) for the first 10 participants at baseline visit (B), mid-point (8 weeks) and endpoint (16 weeks) visits. MD: median difference. (**c**) Severity of Alopecia Tool (SALT) scores for the first seven participants with alopecia areata in the trial. (**d**) Images of participant AA6 at baseline versus week 16. (**e**) Eczema Area and Severity Index (EASI) scores for two participants with mild atopic dermatitis. (**f**) Images of participant AA2 showing improvement in atopic dermatitis upon tofacitinib treatment. p values not shown as per interim analysis plan.

The online version of this article includes the following source data and figure supplement(s) for figure 5:

**Source data 1.** Skin pathology metrics for clinical trial participants.

**Figure supplement 1.** Tofacitinib improves diverse skin pathologies in Down syndrome.

index (PASI), or the eczema area and severity index (EASI; see Materials and methods, see protocol in *Reporting standard 1*). The interim analysis showed that seven of the ten participants had an improvement in the IGA score and eight of the ten reported some improvement on their life quality related to their skin condition as measured by the DLQI (*Figure 5a-b*, *Figure 5—source data 1*). The most striking effects were observed for alopecia areata (*Figure 5c-d*, *Figure 5—figure supplement 1a*). Five of six participants with alopecia areata showed scalp hair regrowth, with the exception being a male participant (AA1) with history of alopecia totalis for 20+years who only showed facial hair and eyelash re-growth. One participant presented with psoriasis due to psoriatic arthritis and experienced an almost complete remission of psoriatic arthritis symptoms (Ps1, *Figure 5—figure supplement 1b–c*). For the two participants that presented with atopic dermatitis, the clinical manifestations were markedly reduced during tofacitinib treatment (*Figure 5e–f*). A total of five participants were affected by HS, three of them as the qualifying condition (HS1-3). No clear trend was seen in the Modified Sartorius Scale (MSS) score used to monitor HS (*Figure 5—figure supplement 1d–e*).

Altogether, these results indicate that JAK inhibition could provide therapeutic benefit for several autoimmune/inflammatory skin conditions more common in DS.

## Tofacitinib normalizes IFN scores and decreases pathogenic cytokines and autoantibodies

It is well demonstrated that individuals with DS display elevated IFN signaling across multiple immune and non-immune cell types (*Sullivan et al., 2016*; *Waugh et al., 2019*; *Araya et al., 2019*; *Powers et al., 2019*). Using an IFN transcriptional score composed of 16 interferon-stimulated genes (ISGs) (*Honda et al., 2006*) measured via bulk RNA sequencing of peripheral blood RNA, individuals with DS in the HTP cohort study show a significant increase in these scores (*Galbraith et al., 2023*; *Figure 6a*). Reduction of IFN scores is designated as a primary endpoint in the trial. At baseline, clinical trial participants show IFN scores within the typical range for DS, but values are decreased at 2, 8, and 16 weeks of tofacitinib treatment (*Figure 6a*, *Figure 6—source data 1*). Time course analysis revealed that most participants show a decrease in IFN scores as soon as two weeks of treatment which is sustained over time, with two clear exceptions (*Figure 6—figure supplement 1a*). At the 8 week study midpoint, 9 of 10 participants had decreased IFN scores relative to baseline, except participant AA2 who reported a COVID-19 vaccination three days prior to the visit and was pausing tofacitinib at the time of the blood draw (*Figure 6—figure supplement 1a*). At the 16 week time point, nine of ten participants had decreased IFN scores, with the exception being AA4, who developed an URI in the week prior to the blood draw (*Figure 6—figure supplement 1a*). Therefore, although all participants displayed decreased IFN scores at one or more time points during the treatment, IFN scores could be sensitive to immune triggers. Analysis of individual ISGs composing the IFN score revealed that whereas many ISGs elevated in DS display reduced expression upon tofacitinib treatment (e.g. *RSAD2*, *IFI44L*), others do not (e.g. *BPGM*) (*Figure 6b*, *Figure 6—figure supplement 1b*). To investigate this further, we defined the impact of tofacitinib on all 136 ISGs significantly elevated in DS that are not encoded on chr21 (*Galbraith et al., 2023*; *Figure 6c*). Collectively, ISGs as a group are significantly downregulated upon tofacitinib treatment, but the effect is not uniform across all ISGs (*Figure 6c*), indicating that JAK1/3 inhibition does not reduce all IFN signaling elevated in DS, which could be explained by the fact that the IFN pathways also employ JAK2 for signal transduction (*Schwartz et al., 2016*; *Schwartz et al., 2017*). Global analysis of transcriptome changes revealed that tofacitinib treatment reverses the dysregulation of many gene signatures observed in DS, effectively attenuating many pro-inflammatory signatures beyond IFN gamma and alpha responses, such as Inflammatory Response, TNF-α signaling via NFkB, IL-2 STAT5 signaling, and IL-6 JAK STAT3 signaling (*Figure 6—figure supplement 1c*). Tofacitinib also reversed elevation of genes involved in Oxidative Phosphorylation and dampened downregulation of gene sets involved in Wnt/Beta Catenin and Hedgehog Signaling (*Figure 6—figure supplement 1c, d*). Conversely, tofacitinib did not rescue elevation of genes involved in Heme Metabolism or Mitotic Spindle (*Figure 6—figure supplement 1c, d*), suggesting that these transcriptome changes are not tied to the inflammatory profile of DS.

A secondary endpoint in the trial is decrease of peripheral inflammatory markers as defined by a composite cytokine score derived from measurements of TNF-α, IL-6, CRP, and IP-10, and which is significantly increased in participants with DS in the HTP study (*Figure 6d*). At baseline, clinical trial participants show cytokine scores within the range observed for DS, but these values decrease at 2, 8

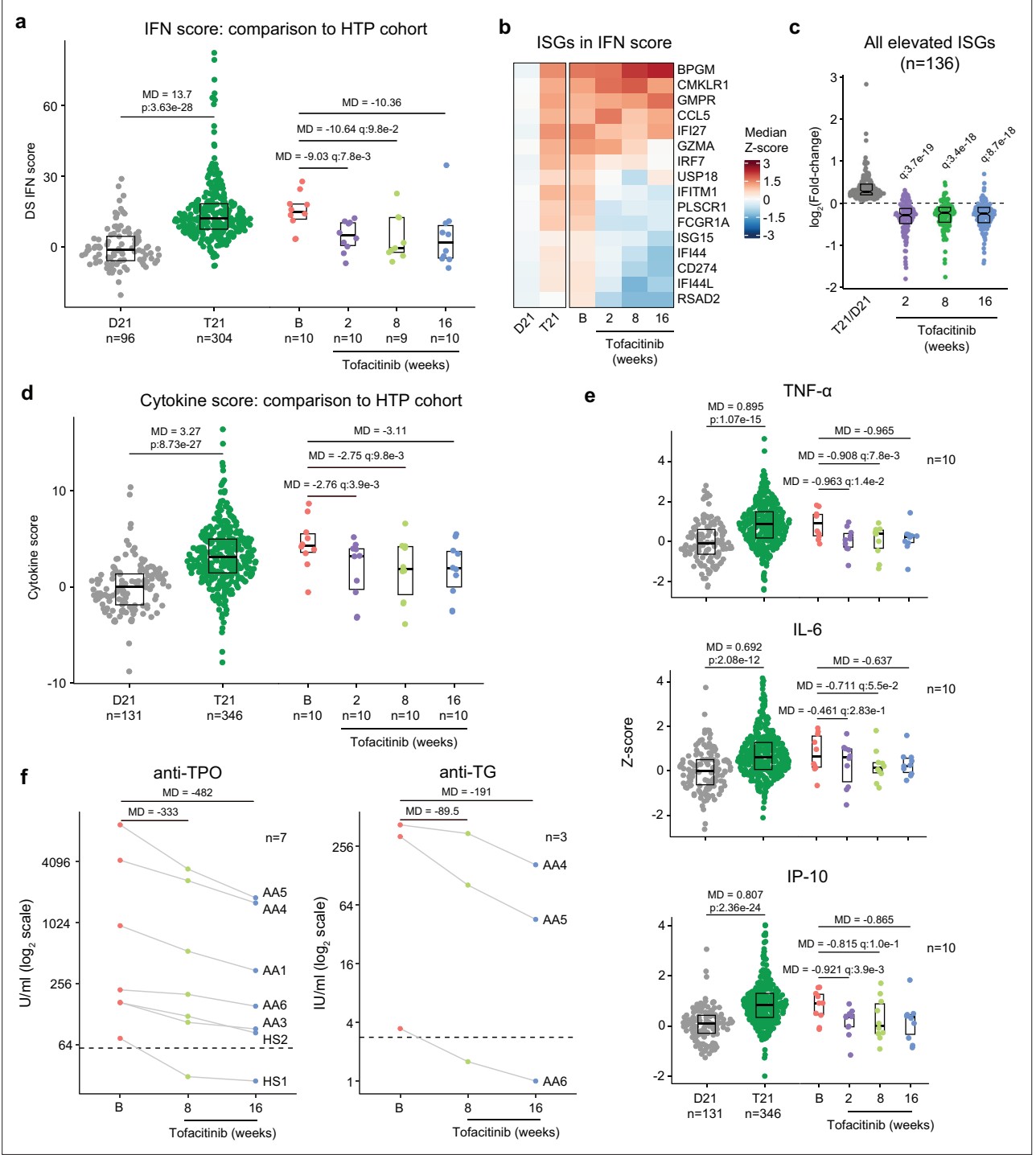

**Figure 6.** Tofacitinib reduces IFN scores, hypercytokinemia, and pathogenic autoantibodies in Down syndrome. (**a**) Comparison of interferon (IFN) transcriptional scores derived from whole blood transcriptome data for research participants in the Human Trisome Project (HTP) cohort study by karyotype status (D21, grey; T21, green) and the clinical trial cohort at baseline (B), and weeks 2, 8, and 16 of tofacitinib treatment. Data are represented as modified sina plots with boxes indicating quartiles. Sample sizes are indicated below the x-axis. Horizontal bars indicate comparisons between groups with median differences (MD) with p-values from Mann-Whitney U-tests (HTP cohort) or q-values from paired Wilcox tests (clinical trial). q value for the 16-week endpoint is not shown as per interim analysis plan. (**b**) Heatmap displaying median z-scores for the indicated groups (as in a) for the 16 interferon-stimulated genes (ISGs) used to calculate IFN scores. (**c**) Analysis of fold changes for 136 ISGs not encoded on chr21 that are significantly elevated in Down syndrome (T21 versus D21) at 2, 8, and 16 weeks of tofacitinib treatment relative to baseline. Sample sizes as in a. q-values above each group indicate significance of Mann-Whitney U-tests against log2-transformed fold-change of 0 (no-chance), after Benjamini-Hochberg correction for multiple testing. (**d**) Comparison of cytokine score distributions for the HTP cohort by karyotype status (D21, T21) versus the clinical trial cohort at

*Figure 6 continued on next page*

*Figure 6 continued*

baseline (B) and 2, 8, and 16 weeks of tofacitinib treatment. Data are represented as modified sina plots with boxes indicating quartiles. Sample sizes are indicated below the x-axis. Horizontal bars indicate comparisons between groups with median differences (MD) with p-values from Mann-Whitney U-tests (HTP cohort) and q-values from paired Wilcox tests (clinical trial). q value for the 16-week endpoint is not shown as per interim analysis plan. (**e**) Comparison of plasma levels of cytokines in the HTP cohort by karyotype status (D21, T21) and the clinical trial cohort at baseline (B) versus 2, 8, and 16 weeks of tofacitinib treatment. Data are represented as modified sina plots with boxes indicating quartiles. Sample sizes are indicated below x-axis. Horizontal bars indicate comparisons between groups with median differences (MD) with p-values from Mann-Whitney U-tests (HTP cohort) and q values from paired Wilcox tests (clinical trial). q value for the 16-week endpoint is not shown as per interim analysis plan. (**f**) Plots showing levels of autoantibodies against thyroid peroxidase (TPO) and thyroglobulin (TG) at baseline versus 8 and 16 weeks of tofacitinib treatment. Sample sizes are indicated in each plot.

The online version of this article includes the following source data and figure supplement(s) for figure 6:

**Source data 1.** Molecular markers of inflammation and autoimmunity in clinical trial participants.

**Figure supplement 1.** JAK inhibition reduces multiple markers of inflammation and autoimmunity in Down syndrome.

and 16 weeks relative to baseline (*Figure 6d–e*, *Figure 6—figure supplement 1e*, *Figure 6—source data 1*). The decreases in TNF-α and IL-6 observed upon tofacitinib treatment indicate that elevation of these potent inflammatory cytokines requires sustained JAK/STAT signaling in DS (*Figure 6e*). As for the IFN scores assessment, time course analysis revealed that most participants show decreases in cytokine scores within two weeks of treatment that are sustained over time, again with the exception of AA2 at week 8 and AA4 at week 16. This reveals a correspondence between RNA-based transcriptional IFN scores and circulating levels of these cytokines in plasma, while also illustrating that both metrics may remain sensitive to immune triggers (*Figure 6—figure supplement 1f–g*).

One tertiary endpoint of the trial investigates the impact of tofacitinib treatment on levels of autoantibodies and markers employed to diagnose AITD [e.g., anti-TPO, anti-TG, anti-thyroid stimulating hormone receptor (TSHR)] and celiac disease [e.g., anti-tissue transglutaminase (tTG), anti-deamidated gliadin peptide (DGP)]. Seven of the 10 participants presented at baseline with anti-TPO levels above the upper limit of normal (ULN, 60 U/mL), and all seven experienced a decrease in these auto-antibodies at 8 weeks and 16 weeks relative to baseline (*Figure 6f*, *Figure 6—source data 1*). In fact, for one participant (HS1) the levels decreased below the ULN while on the trial. All seven of these participants had a history of thyroid disease (*Figure 4c*), which was being medically managed and/or clinically monitored with acceptable TSH and T4 values. Additionally, three of these seven participants also had anti-TG levels above the ULN (4 IU/mL) and all three showed a decrease from baseline levels while on tofacitinib at both 8 and 16 weeks, with one participant (AA6) falling below the ULN upon treatment (*Figure 6f*). Three participants also had anti-TSHr levels above the ULN, but no clear changes were observed upon treatment (*Figure 6—source data 1*). None of the 10 participants displayed anti-tTG or anti-DGP levels detected above ULN at screening.

Altogether, these results indicate that tofacitinib treatment decreases IFN scores, levels of key pathogenic cytokines, and key autoantibodies involved in AITD. Importantly, tofacitinib treatment lowers IFN scores and cytokine levels to within the range observed in the general population, not below, indicating that this immunomodulatory strategy can provide therapeutic benefit in DS without overt immune suppression.

## Discussion

An increasing body of evidence indicates that immune dysregulation contributes to the pathophysiology of DS and that immunomodulatory therapies could provide multidimensional benefits in this population. In mouse models, triplication of four *IFNR* genes contributes to multiple hallmarks of DS (*Maroun et al., 2000*; *Waugh et al., 2023*) and JAK inhibition attenuates global dysregulation of gene expression (*Galbraith et al., 2023*) while rescuing key phenotypes, such as lethal immune hypersensitivity (*Tuttle et al., 2020*) and CHDs (*Chi et al., 2023*). The fact that gene signatures of IFN hyperactivity are present in human embryonic tissues with T21 (*Bhattacharya et al., 2023*) and embryonic tissues from mouse models of DS (*Aziz et al., 2018*; *Waugh et al., 2023*) indicates that the harmful effects of IFN hyperactivity could start in utero, supporting the notion that DS could be understood, in part, as an inborn error of immunity with similarities to monogenic interferonopathies (*Rodero and Crow, 2016*).

Results presented here demonstrate that T21 causes widespread multi-organ autoimmunity of pediatric onset, with production of autoantibodies targeting every major organ system. These results justify additional efforts to define the key pathogenic autoantibodies in DS beyond those commonly associated with AITD and celiac disease. Our analysis found significant associations between specific autoantibodies and some conditions more common in DS, but the diagnostic value of these observations will require validation efforts in much larger cohorts, which could lead to a personalized medicine approach for the management of autoimmunity in DS. For example, we found autoantibodies associated with various forms of auditory dysfunction (*Figure 1g*), suggesting the possibility of autoimmune hearing loss in DS (*Breslin et al., 2020*). Elevated levels of anti-TPO in individuals with history of use of ear tubes suggests an interplay between otitis media and endocrine dysfunction in DS (*Koçyiğit et al., 2017*). For example, it is possible that recurrent ear infections cause a chronic immune stimulus that lead to eventual breach of tolerance in this autoimmunity-prone population, even perhaps through epitope mimicry (*Ercolini and Miller, 2009*). Antibodies targeting MUSK, which we found to be elevated in DS and associated with co-occurring neurological phenotypes (*Figure 1g–h*), have been linked to development of myasthenia gravis, a chronic autoimmune neuromuscular disease that causes weakness in the skeletal muscles (*Dresser et al., 2021*). Whether MUSK antibodies associate with similar phenotypes in DS will require further investigation. Elevation of SRP68 autoantibodies in DS (*Figure 1d and f*), which are common in necrotizing myopathies with cardiovascular involvement (*Allenbach et al., 2020*), suggests a potential autoimmune basis for musculoskeletal and cardiovascular complications in DS, which also warrants additional research.

We observed constitutive global immune remodeling and hypercytokinemia regardless of reported diagnoses of autoimmune disease or measurable autoantibody production from an early age, indicative of an autoimmunity-prone state throughout the lifespan. Although many cytokines elevated in DS have well demonstrated pathogenic roles in the etiology of autoimmune diseases in the general population (e.g. TNF-α, IL6), their consistent upregulation in DS regardless of clinical evidence of autoimmune pathology indicates the existence of a prolonged pre-clinical period, where the hypercytokinemia likely precedes evident tissue damage and symptomology. Alternatively, it is possible that these elevated cytokines are contributing the overall pathophysiology of DS (e.g. cognitive impairments, complications from viral infections) without formal diagnosis of an autoimmune disease. Therefore, measurements of specific immune cell types or cytokines in the bloodstream are unlikely to provide diagnostic value for autoimmunity in DS. However, antigen-specific immune assays, such as T cell or B cell activation assays, may reveal the specific timing of loss of tolerance and transition to clinical phenotypes. Future studies should also include analysis of tissue-resident immune cells, which may identify sites of local autoimmune attack in DS.

Among the many strategies that could be used to attenuate IFN hyperactivity, JAK inhibitors are the most well-studied and have the most approved indications (*Shawky et al., 2022*). Of the more than ten globally-approved JAK inhibitors (*Shawky et al., 2022*), we chose to employ in our clinical trial the JAK1/3 inhibitor tofacitinib, which is used to treat diverse autoimmune/inflammatory conditions and which was approved in 2020 for treatment of polyarticular course juvenile idiopathic arthritis (pcJIA) in children 2 years and older (*Ruperto et al., 2021*; *Shawky et al., 2022*). Notably, all four IFNRs encoded on chr21 utilize JAK1 for signal transduction in combination with either JAK2 or TYK2, making JAK1 inhibitors the most logical choice to dampen the effects of *IFNR* gene triplication. As part of the clinical trial protocol, the approved interim analysis was designed to qualitatively evaluate feasibility and initial safety data on the first 10 participants completing a 16-week course of tofacitinib treatment. This analysis established that there were no AEs that required a change or cessation of tofacitinib dosing and that this medicine is well tolerated in individuals with DS. The clear benefits observed for diverse autoimmune skin conditions align with an increasing body of evidence supporting the use of JAK inhibition for immunodermatological conditions, including their recent approval for alopecia areata and atopic dermatitis in the general population (*King and Craiglow, 2023*; *Tampa et al., 2023*). At this sample size, the effects of tofacitinib on HS are inconclusive. Although some participants and caregivers reported benefits in terms of fewer flares and of lesser severity, the MSS metric did not show a clear trend, which may reveal the need for more frequent or different types of monitoring for HS, a condition that cycles periodically in severity.

Our results indicate that tofacitinib does not fully suppress the immune response in people with DS, but rather attenuates IFN scores and cytokine scores to levels observed in the general

population, which is an important consideration given the likely requirement for long-term use of the drug in this population. Furthermore, the effects of the drug are clearly gene-specific, highlighting the presence of inflammatory processes that may not be attenuated with this inhibitor, which could be beneficial in terms of preserving immune activity. Importantly, during treatment, both IFN scores and cytokine scores remain sensitive to immune stimuli, as evidenced by participants who had received a vaccine or experienced an URI before a blood draw (*Figure 6a*, *Figure 6— figure supplement 1f*). Overall, it is encouraging that key inflammatory markers decreased in a relatively short timeframe, likely offering systemic benefits beyond skin pathology. Importantly, the fact that levels of IL-6 and TNF-α are reduced upon tofacitinib treatment supports the use of JAK inhibitors over TNF-blockers or anti-IL-6 agents in this population. Although TNF-α-blockers are recommended to be used first in the treatment of rheumatoid arthritis in the general population (*Ytterberg et al., 2022*), the value of this recommendation in people with DS remains to be defined. The clear decrease in anti-TPO and anti-TG levels indicates that autoreactive B cell function requires elevated JAK/STAT signaling, but whether this effect is cell-autonomous versus a consequence of a reduced systemic inflammatory milieu will require further investigation. Defining the effect of tofacitinib on other autoantibodies elevated in DS will also require a larger sample size and may be revealed in the full dataset after completion of this trial, along with analysis of potential remodeling of the B cell lineage upon JAK inhibition, such as effects on mature B cells and plasmablast populations.

Lastly, this ongoing clinical trial includes measurements of various dimensions of neurological function not reported here. Although the absence of a placebo control arm may impede a clear interpretation of any effect of JAK inhibition on cognitive function, preliminary results have prompted the design and launch of a second trial (NCT05662228) aimed at defining the relative safety and efficacy of tofacitinib, intravenous immunoglobulin (IVIG), and the benzodiazepine lorazepam for Down syndrome Regression Disorder (DSRD), a condition characterized by sudden loss of neurological function (*Santoro et al., 2022*; *Rachubinski et al., 2024*).

Altogether, these findings justify both a deeper investigation of all the deleterious effects of autoimmunity and hyperinflammation in DS and the expanded testing of immunomodulatory strategies for diverse aspects of DS pathophysiology, even perhaps from an early age.

## Acknowledgements

This work was supported primarily by NIH grant R61AR077495. Additional funding was provided by NIH grants R01AI150305 (JME), T32CA190216 (KAW), 2T32AR007411-31 (KAW), UM1TR004399 (data generation and REDCap support), P30CA046934 (support of shared resources), the Linda Crnic Institute for Down Syndrome, the Global Down Syndrome Foundation, the Anna and John J Sie Foundation, the Human Immunology and Immunotherapy Initiative, the University of Colorado School of Medicine, the Boettcher Foundation, and Fast Grants. We are grateful to all research participants and their families involved in the Human Trisome Project and the clinical trial. We thank Lyndy Bush for administrative support, Dr. Kim Jordan and her team at the Human Immune Monitoring Shared Resource for outstanding service in generation of the immune marker dataset, and Dr. Eric Clambey and his team at the Flow Cytometry Shared Resource for outstanding service in generation of the mass cytometry dataset. We are also grateful to the Colorado Translational and Sciences Institute and the Colorado Multiple Institutional Review Board for assistance in all clinical research projects involving the Crnic Institute. Special thanks to Michelle Sie Whitten, the team at the Global Down Syndrome Foundation, Dr. John Reilly, and Dr. Ron Sokol for logistical support at multiple stages of the project.

## Additional information

### Competing interests

Joaquín M Espinosa: has provided consulting services for Eli Lilly Co, Gilead Sciences Inc, and Biohaven Pharmaceuticals and serves on the advisory board of Perha Pharmaceuticals. The other authors declare that no competing interests exist.

## Funding

| Funder | Grant reference number | Author |
| --- | --- | --- |
| National Institute of Arthritis and Musculoskeletal and Skin Diseases | R61AR077495 | Angela L Rachubinski Emily Gurnee Cory A Dunnick David A Norris Joaquín M Espinosa |
| National Institute of Allergy and Infectious Diseases | R01AI150305 | Joaquín M Espinosa |
| National Cancer Institute | T32CA190216 | Katherine A Waugh |
| National Institute of Arthritis and Musculoskeletal and Skin Diseases | 2T32AR007411-31 | Katherine A Waugh |
| National Center for Advancing Translational Sciences | UM1TR004399 | Joaquín M Espinosa |
| National Cancer Institute | P30CA046934 | Joaquín M Espinosa |
| Global Down Syndrome Foundation | | Joaquín M Espinosa |
| Anna and John J. Sie Foundation | | Joaquín M Espinosa |
| Boettcher Foundation | | Kelly D Sullivan |
| Fast Grants | | Joaquín M Espinosa |

The funders had no role in study design, data collection and interpretation, or the decision to submit the work for publication.

## Author contributions

Angela L Rachubinski, Data curation, Formal analysis, Supervision, Funding acquisition, Investigation, Visualization, Methodology, Writing – original draft, Project administration; Elizabeth Wallace, Emily Gurnee, Keith P Smith, Katherine A Waugh, Ross E Granrath, Eleanor Britton, Hannah R Lyford, Investigation, Methodology, Writing – review and editing; Belinda A Enriquez-Estrada, Kayleigh R Worek, Data curation, Investigation, Methodology, Writing – review and editing; Paula Araya, Conceptualization, Data curation, Formal analysis, Investigation, Visualization, Methodology, Writing – review and editing; Micah G Donovan, Neetha Paul Eduthan, Data curation, Formal analysis, Investigation, Visualization, Writing – review and editing; Amanda A Hill, David A Norris, Supervision, Funding acquisition, Investigation, Methodology, Project administration, Writing – review and editing; Barry Martin, Investigation, Writing – review and editing; Kelly D Sullivan, Formal analysis, Supervision, Investigation, Visualization, Writing – review and editing; Lina Patel, Data curation, Formal analysis, Supervision, Investigation, Methodology, Writing – review and editing; Deborah J Fidler, Supervision, Investigation, Methodology, Writing – review and editing; Matthew D Galbraith, Data curation, Software, Formal analysis, Supervision, Investigation, Visualization, Methodology, Writing – original draft, Writing – review and editing; Cory A Dunnick, Conceptualization, Formal analysis, Supervision, Funding acquisition, Investigation, Methodology, Writing – review and editing; Joaquín M Espinosa, Conceptualization, Formal analysis, Supervision, Funding acquisition, Investigation, Visualization, Methodology, Writing – original draft, Project administration, Writing – review and editing

## Author ORCIDs

Kelly D Sullivan https://orcid.org/0000-0003-2725-0205
Matthew D Galbraith https://orcid.org/0000-0003-0485-3927
Joaquín M Espinosa https://orcid.org/0000-0001-9048-1941

## Ethics

registration NCT04246372.
All aspects of this study were conducted in accordance with the Declaration of Helsinki under human research protocols approved by the Colorado Multiple Institutional Review Board (COMIRB): protocol

#15-2170 - The Human Trisome Project (NCT02864108) and protocol #19-1362 - Safety and efficacy of tofacitinib for immune skin conditions in Down syndrome (NCT04246372). Informed consent, including consent to publish results and data sharing, was obtained from research participants or their legally authorized representatives.

Reviewer #1 (Public review): https://doi.org/10.7554/eLife.99323.3.sa1
Reviewer #2 (Public review): https://doi.org/10.7554/eLife.99323.3.sa2
Reviewer #3 (Public review): https://doi.org/10.7554/eLife.99323.3.sa3
Author response https://doi.org/10.7554/eLife.99323.3.sa4

---

## Additional files

### Supplementary files
Supplementary file 1. Cohort characteristics for participants in the Human Trisome Project involved in this study and for subsets of this cohort that were included in specific analyses.

Supplementary file 2. Characteristics of clinical trial cohort. (A) Minimum qualifying scores for skin conditions. (B) Cohort characteristics for clinical trial participants.

MDAR checklist

Reporting standard 1. Clinical trial.

### Data availability
Demographic and health history data for research participants in the Human Trisome Project study are available on both the Synapse data sharing platform (https://doi.org/10.7303/syn31488784) and through the INCLUDE Data Hub (https://portal.includedcc.org/). Mass cytometry data for 380+ HTP research participants are available in Synapse (https://doi.org/10.7303/syn53185253). Targeted plasma proteomics for inflammatory markers using Meso Scale Discovery (MSD) assays for 470+ HTP research participants can be accessed through Synapse (https://doi.org/10.7303/syn31475487) and the INCLUDE Data Hub. Whole blood transcriptome data for 400 HTP research participants can be accessed through NCBI Gene Expression Omnibus (GSE190125). Whole blood transcriptome data for 10 clinical trial participants at baseline and after 2, 8, and 16 weeks of tofacitinib treatment can be accessed NCBI Gene Expression Omnibus (GSE251967). Targeted plasma proteomics for inflammatory markers using Meso Scale Discovery (MSD) assays for 10 clinical trial participants can be accessed through Synapse (https://doi.org/10.7303/syn53185252).

The following datasets were generated:

| Author(s) | Year | Dataset title | Dataset URL | Database and Identifier |
|---|---|---|---|---|
| Galbraith MD, Rachubinski AL, Espinosa JM | 2024 | Crnic Institute Human Trisome Project - JAK inhibition in Down syndrome: PolyA RNAseq from whole blood | https://www.ncbi.nlm.nih.gov/geo/query/acc.cgi?acc=GSE251967 | NCBI Gene Expression Omnibus, GSE251967 |
| Galbraith MD, Rachubinski AL, Espinosa JM | 2024 | Tofacitinib for Immune Skin Conditions in DS - Plasma inflammatory markers | https://doi.org/10.7303/syn53185252 | Synapse, 10.7303/syn53185252 |

The following previously published datasets were used:

| Author(s) | Year | Dataset title | Dataset URL | Database and Identifier |
|---|---|---|---|---|
| Galbraith MD, Rachubinski AL, Espinosa JM | 2022 | Human Trisome Project Sample-Participant metadata | https://doi.org/10.7303/syn31488784 | Synapse, 10.7303/syn31488784 |

*Continued on next page*

*Continued*

| Author(s) | Year | Dataset title | Dataset URL | Database and Identifier |
|---|---|---|---|---|
| Galbraith MD, Araya P, Waugh K, Rachubinski AL, Espinosa JM | 2023 | Human Trisome Project Mass cytometry v2 (CyTOF) | https://doi.org/10.7303/syn53185253 | Synapse, 10.7303/syn53185253 |
| Galbraith MD, Rachubinski AL, Espinosa JM | 2022 | Human Trisome Project Plasma inflammatory markers (Multiplex Immunoassay) | https://doi.org/10.7303/syn31475487 | Synapse, 10.7303/syn31475487 |
| Galbraith MD, Rachubinski AL, Smith KP, Sullivan KD, Espinosa JM | 2023 | Crnic Institute Human Trisome Project: PolyA RNA-seq from whole blood | https://www.ncbi.nlm.nih.gov/geo/query/acc.cgi?acc=GSE190125 | NCBI Gene Expression Omnibus, GSE190125 |

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

# Appendix 1

**Appendix 1—key resources table**

| Reagent type (species) or resource | Designation | Source or reference | Identifiers | Additional information |
|---|---|---|---|---|
| Antibody | Mouse monoclonal anti-human CD11c (clone Bu15) | Fluidigm | Cat # 3147008; RRID:AB_2687850 | Lot 3431914, 1:100 |
| Antibody | Mouse monoclonal anti-human CD123 (clone 6 H6) | Fluidigm | Cat # 3143014B; RRID:AB_2811081 | Lot 3431917, 1:100 |
| Antibody | Mouse monoclonal anti-human CD127 (clone A019D5) | Fluidigm | Cat # 3149011; RRID:AB_2661792 | Lot 3321819, 1:100 |
| Antibody | Mouse monoclonal anti-human CD14 (clone M5E2) | Fluidigm | Cat # 3151009B; RRID:AB_2810244 | Lot 2191914, 1:100 |
| Antibody | Mouse monoclonal anti-human CD15 (Clone W6D3) | BioLegend | Cat # 323002; RRID:AB_756008 | Lot B254011, 1:67 |
| Antibody | Mouse monoclonal anti-human CD16 (clone B73.1) | BioLegend | Cat # 360702; RRID:AB_2562693 | Lot B243320, 1:33 |
| Antibody | Mouse monoclonal anti-human CD161 (clone DX12) | BD Biosciences | Cat # 556079; RRID:AB_396346 | Lot 9115548, 1:33 |
| Antibody | Mouse monoclonal anti-human CD19 (clone HIP19) | Fluidigm | Cat # 3142001; RRID:AB_2651155 | Lot 3031906, 1:100 |
| Antibody | Mouse monoclonal anti-human CD1c (clone L161) | BioLegend | Cat # 331501; RRID:AB_1088996 | Lot B265380, 1:100 |
| Antibody | Mouse monoclonal anti-human CD25 (clone 2 A3) | Fluidigm | Cat # 3169003; RRID:AB_2661806 | Lot 0342004, 1:100 |
| Antibody | Mouse monoclonal anti-human CD27 (clone L128) | Fluidigm | Cat # 3167006B; RRID:AB_2811093 | Lot 2851804, 1:400 |
| Antibody | Mouse monoclonal anti-human CD279/PD1 (clone EH12.2H7) | Fluidigm | Cat # 3155009B; RRID:AB_2811087 | Lot 2971910, 1:133 |
| Antibody | Mouse monoclonal anti-human CD3 (clone UCHT1) | DVS Sciences | Cat # 3154003B; RRID:AB_2811086 | Lot 0071917, 1:100 |
| Antibody | Mouse monoclonal anti-human CD33 (clone WM53) | BioLegend | Cat # 303402; RRID:AB_314346 | Lot B277151, 1:33 |
| Antibody | Mouse monoclonal anti-human CD34 (clone 581) | Fluidigm | Cat # 3163014B; RRID:AB_2811091 | Lot 2651705, 1:33 |
| Antibody | Mouse monoclonal anti-human CD38 (clone HIT2) | Fluidigm | Cat # 3172007B; RRID:AB_2756288 | Lot 0861906, 1:100 |
| Antibody | Mouse monoclonal anti-human CD4 (clone RPA-T4) | Fluidigm | Cat # 3145001; RRID:AB_2661789 | Lot 2681902, 1:100 |
| Antibody | Mouse monoclonal Anti-Human CD45 (Clone HI30) | Fluidigm | Cat # 3089003B; RRID:AB_2661851 | Lot 2801911, 1:100 |
| Antibody | Mouse monoclonal anti-human CD45RA (clone HI100) | BioLegend | Cat # 304102; RRID:AB_314406 | Lots B295482, B255475, 1:33 |
| Antibody | Mouse monoclonal anti-human CD45RO (clone UCHL1) | Fluidigm | Cat # 3164007B; RRID:AB_2811092 | Lot 2431806, 1:100 |
| Antibody | Mouse monoclonal anti-human CD56 (clone N901) | Fluidigm | Cat # 3176009B; RRID:AB_2811096 | Lot 3171701, 1:50 |
| Antibody | Mouse monoclonal anti-human CD7 (clone CD7-6B7) | DVS Sciences | Cat # 3153014B; RRID:AB_2811084 | Lot 0282010, 1:100 |
| Antibody | Mouse monoclonal anti-human CD8a (clone RPA-T8) | Fluidigm | Cat # 3162015; RRID:AB_2661802 | Lot 0171813, 1:100 |

*Appendix 1 Continued on next page*

*Appendix 1 Continued*

| Reagent type (species) or resource | Designation | Source or reference | Identifiers | Additional information |
|---|---|---|---|---|
| Antibody | Mouse monoclonal anti-human CD95 (clone DX2) | BioLegend | Cat # 305602; RRID:AB_314540 | Lot B241963, 1:67 |
| Antibody | Mouse monoclonal anti-human HLA-DR (clone L243) | Fluidigm | Cat # 3174001B; RRID:AB_2665397 | Lot 0991901, 1:100 |
| Antibody | Mouse monoclonal anti-human IgD (clone IA6-2) | Fluidigm | Cat # 3146005B; RRID:AB_2811082 | Lot 2561908, 1:100 |
| Antibody | Mouse monoclonal anti-human IgM (clone MHM-88) | BioLegend | Cat # 314502; RRID:AB_493003 | Lot B264164, 1:33 |
| Antibody | Mouse monoclonal anti-human PD-L1 (clone 29E.2A3) | Fluidigm | Cat # 3156026; RRID:AB_2687855 | Lot 2761903, 1:100 |
| Antibody | Mouse monoclonal anti-human PICP (Clone PCIDG10) | Millipore | Cat # MAB1913; RRID:AB_94406 | Lots 3328869, 3389939, 1:133 |
| Antibody | Mouse monoclonal anti-human EMR1 (Clone BM8) | BioLegend | Cat # 123102; RRID:AB_893506 | Lot B264265, 1:33 |
| Antibody | Mouse monoclonal anti-human TCR Va7.2 (Clone 3 C10) | BioLegend | Cat # 351702; RRID:AB_10900258 | Lots B282453, 1:33 |
| Antibody | Mouse monoclonal anti-human FOXP3 (clone 259D/C7) | Fluidigm | Cat # 3159028 A; RRID:AB_2811088 | Lots 1812006, 2631804, 1:50 |
| Antibody | Rabbit monoclonal anti-human phospho-4E-BP1 (Thr37/Thr46) (clone 236B4) | Cell Signaling Technology | Cat # 2855; RRID:AB_560835 | Lots 29, 31, 1:20 |
| Antibody | Rabbit monoclonal anti-human phospho-STAT1 (Tyr701) (clone 58D6) | Cell Signaling Technology | Cat # 9167; RRID:AB_561284 | Lot 22, 1:400 |
| Antibody | Mouse monoclonal anti-human GZMB (clone GB11) | Fluidigm | Cat # 3173006B; RRID:AB_2811095 | Lot 1611909, 1:100 |
| Antibody | Mouse monoclonal anti-human CD11b (Clone ICRF44) | BioLegend | Cat # 301302; RRID:AB_314154 | Lot B286270, 1:33 |
| Antibody | Mouse monoclonal anti-human TCRgd (Clone 11 F2) | BioLegend | Cat # 331202; RRID:AB_1089222 | Lot B271574, 1:33 |
| Antibody | Mouse monoclonal anti-human Cleaved PARP (Clone F21-852) | BD Pharmingen Customs | Cat # 624084; RRID:NA | Lot 9326323, 1:33 |
| Antibody | Mouse monoclonal anti-human RORgt (Clone 4F3-3C8-2B7) | BioLegend | Cat # 644902; RRID:AB_1595502 | NA, 1:33 |
| Antibody | Mouse monoclonal anti-human T-bet (Clone 4B10) | BioLegend | Cat # 644802; RRID:AB_2810251 | Lot B335065, 1:33 |
| Antibody | Mouse monocloncal anti-human CD66b (Clone G10f5) | BioLegend | Cat # 305102; RRID:AB_314494 | Lot B298277, 1:308 |
| Commercial assay or kit | PAXgene Blood RNA tubes | Qiagen | Cat # 762165 | |
| Commercial assay or kit | PAXgene Blood RNA Kit | Qiagen | Cat # 762164 | |
| Commercial assay or kit | Allprep DNA/RNA/miRNA Universal Kit | Qiagen | Cat # 80224 | |
| Commercial assay or kit | GlobinClear kit | ThermoFisher Scientific | Cat # AM1980 | |
| Commercial assay or kit | NEBNext Poly(A) mRNA Magnetic Isolation Module | New England Biolabs | Cat # E7490 | |
| Commercial assay or kit | NEBNext Ultra II Directional RNA Library Prep Kit for Illumina | New England Biolabs | Cat # E7760; | |

*Appendix 1 Continued on next page*

*Appendix 1 Continued*

| Reagent type (species) or resource | Designation | Source or reference | Identifiers | Additional information |
|---|---|---|---|---|
| Commercial assay or kit | V-PLEX Human Biomarker 54-Plex | MesoScale Discovery | Cat # K15248D | |
| Commercial assay or kit | Transcription Factor Phospho Buffer Set | BD Pharmingen | Cat # 563239 | |
| Commercial assay or kit | Cell Staining Buffer | Fluidigm | Cat # 201068 | |
| Commercial assay or kit | Cell-IDTM 20- Plex Pd Barcoding Kit | Fluidigm | Cat # PRD023 | |
| Commercial assay or kit | Cell-ID Intercalator-Ir | Fluidigm | Cat # 201192 A | |
| Commercial assay or kit | Maxpar Antibody Labeling Kit | Fluidigm | Cat # 201160B | |
| Software, algorithm | R | R Foundation for Statistical Computing | v4.3.1; RRID:SCR_001905 | |
| Software, algorithm | R Studio | R Studio, Inc | v2023.09.1+494; RRID:SCR_000432 | |
| Software, algorithm | Bioconductor | Bioconductor | v3.17; RRID:SCR_006442 | |
| Software, algorithm | Tidyverse collection of packages for R | CRAN | RRID:SCR_019186 | |
| Software, algorithm | limma package for R | Bioconductor | v3.56.2; RRID:SCR_010943 | |
| Software, algorithm | FASTQC | Babraham Institute | v0.11.5; RRID:SCR_014583 | |
| Software, algorithm | FastQ Screen | Babraham Institute | v0.11.0; RRID:SCR_000141 | |
| Software, algorithm | bbduk/BBTools | *Bushnell et al., 2017* | v37.99; RRID:SCR_016968 | |
| Software, algorithm | fastq-mcf/ea-utils | N/A | v1.05; RRID:SCR_005553 | |
| Software, algorithm | HISAT2 | *Kim et al., 2019* | v2.1.0; RRID:SCR_015530 | |
| Software, algorithm | Human genome sequence primary assembly fasta | Gencode | GRCh38; RRID:SCR_014966 | |
| Software, algorithm | Human genome basic annotation GTF file | Gencode | v33; RRID:SCR_014966 | |
| Software, algorithm | Samtools | N/A | v1.5; RRID:SCR_002105 | |
| Software, algorithm | HTSeq-count | N/A | v0.6.1; RRID:SCR_005514 | |
| Software, algorithm | DESeq2 package for R | Bioconductor | v1.28.1; RRID:SCR_015687 | |
| Software, algorithm | fgsea package for R | Bioconductor | v1.26.0; RRID:SCR_020938 | |
| Software, algorithm | Hmisc package for R | CRAN | V5.1.1; RRID:SCR_022497 | |
| Software, algorithm | ggplot2 package for R | CRAN | v3.4.4; RRID:SCR_014601 | |
| Software, algorithm | rstatix package for R | CRAN | v0.7.2; RRID:SCR_021240 | |
| Software, algorithm | ComplexHeatmap package for R | CRAN | v2.4.2; RRID:SCR_017270 | |
| Software, algorithm | tidyheatmap package for R | CRAN | V1.8.1 | |
| Software, algorithm | ggforce package for R | CRAN | v0.4.1 | |
| Software, algorithm | CellEngine | CellCarta, Montreal, Canada | RRID:SCR_022484 | |
| Software, algorithm | flowCore package for R | *Hahne et al., 2009*; Bioconductor | v2.0.1; RRID:SCR_002205 | |
| Software, algorithm | CATALYST package for R | *Chevrier et al., 2018*; Bioconductor | v1.12.2; RRID:SCR_017127 | |

*Appendix 1 Continued on next page*

*Appendix 1 Continued*

| Reagent type (species) or resource | Designation | Source or reference | Identifiers | Additional information |
|---|---|---|---|---|
| Software, algorithm | FlowSOM package for R | *Van Gassen et al., 2015* | v1.20.0; RRID:SCR_016899 | |
| Software, algorithm | ConsensusClusterPlus package for R | Bioconductor; *Wilkerson and Waltman, 2010* | v1.52.0; RRID:SCR_016954 | |
| Software, algorithm | tidySingleCellExperiment package for R | Bioconductor | v1.3.3; RRID:SCR_022493 | |
| Software, algorithm | MEM package for R | *Diggins et al., 2017* | v3; RRID:SCR_022495 | |
| Software, algorithm | Betareg package for R | CRAN; *Cribari-Neto et al., 2021* | v3.1–4; RRID:SCR_022494 | |
| Software, algorithm | Ggeffects package for R | CRAN; *Lüdecke et al., 2021* | v1.1.0; RRID:SCR_022496 | |
| Software, algorithm | cluster package for R | CRAN | v2.1.0 | |
| Software, algorithm | janitor package for R | CRAN | v2.0.1 | |

