## [Editor Report · eLife Assessment]

Rachubinski and colleagues provide an **important** manuscript that includes two major advances in understanding immune dysregulation in a large cohort of individuals with Down syndrome. The work comprises **compelling**, comprehensive, and state-of-the-art clinical, immunological, and autoantibody assessment of autoimmune/inflammatory manifestations. Additionally, the authors report promising results from a clinical trial with the JAK inhibitor tofacitinib for individuals with dermatological autoimmune disease.

---

## [Referee Report · Reviewer #1 (Public review)]

Summary:

This paper represents a huge amount of work on a condition whose patients' health and well-being have not always been prioritized, and only relatively recently has the immune dysregulation seen in patients with Down Syndrome (DS) been garnering major research interest.

This paper provides an unparalleled examination of immune disorder in patients with DS. The authors also report the results from a clinical trial with the JAK inhibitor tofacitinib in DS patients.

Strengths:

This manuscript report an herculean effort and provides an unparalleled examination of immune disorder in a large number of patients with DS.

Weaknesses:

Not a major weakness but, apart from finding an elevation of CD4 T central memory cells and more differentiated plasmablast, several of the alteration reported in this manuscript had already been suggested by a few case reports and very small series. On the other hand, the number of patients (and controls) utilized for this study is remarkable and allows to draw much firmer conclusions.

Comments on revised version:

I don't have any further comments.

---

## [Referee Report · Reviewer #2 (Public review)]

In this manuscript, Rachubinski and colleagues provide a comprehensive clinical, immunological, and autoantibody assessment of autoimmune/inflammatory manifestations of patients with Down syndrome (DS) in a large number of patients with this disorder. These analyses confirm prior results of excess interferon and cytokine signals in DS patients and extend these observations to highlight early-onset immunological aberrancies, far before symptoms occur, as well as characterizing novel autoantibody reactivities in this patient population. Then, the authors report the interim analysis of an open label, Phase II, clinical trial of the JAK1/3 inhibitor, tofacitinib, that aims to define the safety, clinical efficacy, and immunological outcomes of DS patients who suffer from inflammatory conditions of the skin. The clinical trial analysis indicates that the treatment is tolerated without serious adverse effects and that the majority of patients have experienced clinical improvement or remission in their corresponding clinical cutaneous manifestations as well as improvement or normalization of aberrant immunological signals such as cytokines.

The major strength of the study is the recruitment and uniform, systematic evaluation of an impressive number of DS patients. Moreover, the promising early results from the tofacitinib clinical trial pave the way for analysis of a larger number of patients within the Phase II trial and otherwise, which may lead to improved clinical outcomes of affected patients. An inherent weakness of such studies is the descriptive nature of several parameters and the relatively small size of tofacitinib-treated DS patients. However, the descriptive nature of some of the correlative research analyses are of scientific interest and are useful to generate hypotheses for future additional (including mechanistic) work and treatment of 10 DS patients in a formal clinical trial at interim analysis is not a trivial task for a disease like this. The manuscript achieves the aims of the authors and the results support their conclusions. The authors appropriately acknowledge areas that require more research and areas that are not well understood. The results are represented in a useful manner and statistical methods and analyses appear sound.

Comments on revised version:

The authors have satisfactorily addressed my comments in the revised manuscript.

---

## [Referee Report · Reviewer #3 (Public review)]

Summary:

Individuals with Down syndrome (DS) have high rates of autoimmunity and can have exaggerated immune responses to infection that can unfortunately cause significant medical complications. Prior studies from these authors and others have convincingly demonstrated that individuals with DS have immune dysregulation including increased Type I IFN activity, elevated production of inflammatory cytokines (hypercytokinemia), increased autoantibodies, and populations of dysregulated adaptive immune cells that pre-dispose to autoimmunity. Prior studies have demonstrated that using JAK inhibitors to treat patient samples in vitro, in small case series of patients, and in mouse models of DS leads to improvement of immune phenotype and/or clinical disease. This manuscript provides two major advances in our understanding of the immune dysregulation and therapy for patients. First, they perform deep immune phenotyping on several hundred individuals with DS and demonstrate that immune dysregulation is present from infancy. Second, they report promising interim analysis of a Phase II clinical trial of a JAK inhibitor in 10 people with DS and moderate to severe skin autoimmunity.

Strengths and weaknesses:

The relatively large cohort and careful clinical annotation here provides new insights into the immune phenotype of patients with DS. For example, it is interesting that regardless of autoimmune disease or autoantibody status, individuals with DS have elevated cytokines and CRP. Analysis of the cohorts by age demonstrated that some cytokines are significant elevated in people with DS starting in infancy (e.g., IL-9 and IL-17C). Nearly all adults with DS in this study had autoantibodies (98%) and most had six or more autoantibodies (63%), which differed significantly from euploid study participants. This implies that all patients with DS might benefit from early intervention with therapy to reduce inflammation. However, it is also worth considering that an alternative interpretation that since hypercytokinemia does not vary based on disease state in individuals with DS, that this may not be a key factor driving autoimmunity (although it may be relevant for other clinical symptoms such as neuroinflammation).

Small case series have suggested the benefit of JAK inhibitors to treat autoimmunity in DS. This is the first report of a prospective clinical trial to test a JAK inhibitor in this setting. The clinical trial entry criteria included moderate to severe autoimmune skin disease in patients aged 12-50 years with DS, and treatment was with the JAK1/3 inhibitor tofacitinib. This clinical trial is a critically important step for the field. The early results support that treatment is well tolerated with improvement of interferon scores in patients and reduction of autoantibodies. Most patients experienced clinical improvement, with alopecia areata having the greatest response. Treatment may not affect all skin disease equally, for example of the 5 patients with hidradenitis suppurativa, only 1 showed clinical improvement based on skin score. While very promising, the clinical trial results reported here are preliminary and based on interim analysis of 10 patients at 16 weeks. Individuals with DS have a lifelong risk of immune dysregulation and thus it is unclear how long therapy, if of benefit, would need to be continued. Results of longer-term therapy will be informative when considering the risks/benefits of this therapy.

Comments on revised version:

The authors have made appropriate revisions to this important contribution to the literature.

---

## [Author Response]

The following is the authors’ response to the original reviews.

**Public Reviews:**

**Reviewer #1 (Public Review):**
Summary:This paper represents a huge amount of work on a condition whose patients' health and well-being have not always been prioritized, and only relatively recently has the immune dysregulation seen in patients with Down Syndrome (DS) been garnering major research interest.This paper provides an unparalleled examination of immune disorders in patients with DS. The authors also report the results from a clinical trial with the JAK inhibitor tofacitinib in DS patients.Strengths:This manuscript reports a herculean effort and provides an unparalleled examination of immune disorders in a large number of patients with DS.Weaknesses:Not a major weakness but, apart from finding an elevation of CD4 T central memory cells and more differentiated plasmablast, several of the alterations reported in this manuscript had already been suggested by a few case reports and a very small series. On the other hand, the number of patients (and controls) utilized for this study is remarkable and allows for drawing much firmer conclusions.

We are grateful for the Reviewer’s very positive assessment of the work and results presented in this manuscript. We agree that many of the changes in the peripheral immune system reported here had been previously documented by our team and others using smaller sample sizes. However, as the Reviewer appreciated, this study involves an order of magnitude more research participants than previous studies (i.e., ~400 total participants, ~300 of them with trisomy 21 versus ~100 controls), which enabled us to investigate associations between immune changes and clinical variables, while also helping us draw much firmer conclusions.

**Reviewer #2 (Public Review):**
In this manuscript, Rachubinski and colleagues provide a comprehensive clinical, immunological, and autoantibody assessment of autoimmune/inflammatory manifestations of patients with Down syndrome (DS) in a large number of patients with this disorder. These analyses confirm prior results of excess interferon and cytokine signals in DS patients and extend these observations to highlight early-onset immunological aberrancies, far before symptoms occur, as well as characterizing novel autoantibody reactivities in this patient population. Then, the authors report the interim analysis of an open-label, Phase II, clinical trial of the JAK1/3 inhibitor, tofacitinib, that aims to define the safety, clinical efficacy, and immunological outcomes of DS patients who suffer from inflammatory conditions of the skin. The clinical trial analysis indicates that the treatment is tolerated without serious adverse effects and that the majority of patients have experienced clinical improvement or remission in their corresponding clinical cutaneous manifestations as well as improvement or normalization of aberrant immunological signals such as cytokines.The major strength of the study is the recruitment and uniform, systematic evaluation of an impressive number of DS patients. Moreover, the promising early results from the tofacitinib clinical trial pave the way for analysis of a larger number of patients within the Phase II trial and otherwise, which may lead to improved clinical outcomes for affected patients. An inherent weakness of such studies is the descriptive nature of several parameters and the relatively small size of tofacitinib-treated DS patients. However, the descriptive nature of some of the correlative research analyses is of scientific interest and is useful to generate hypotheses for future additional (including mechanistic) work, and treatment of 10 DS patients in a formal clinical trial at interim analysis is not a trivial task for a disease like this. The manuscript achieves the aims of the authors and the results support their conclusions. The authors appropriately acknowledge areas that require more research and areas that are not well understood. The results are represented in a useful manner and statistical methods and analyses appear sound.

We appreciate the very positive evaluation by this Reviewer. We agree with the Reviewer on the descriptive nature of many of the analyses completed and on the value of a larger cohort of individuals with Down syndrome treated with a JAK inhibitor. The clinical trial will involve a total of 40 participants, and we look forward to reporting the results from the full cohort in the near future.

**Reviewer #3 (Public Review):**
Summary:Individuals with Down syndrome (DS) have high rates of autoimmunity and can have exaggerated immune responses to infection that can unfortunately cause significant medical complications. Prior studies from these authors and others have convincingly demonstrated that individuals with DS have immune dysregulation including increased Type I IFN activity, elevated production of inflammatory cytokines (hypercytokinemia), increased autoantibodies, and populations of dysregulated adaptive immune cells that pre-dispose to autoimmunity. Prior studies have demonstrated that using JAK inhibitors to treat patient samples in vitro, in small case series of patients, and in mouse models of DS leads to improvement of immune phenotype and/or clinical disease. This manuscript provides two major advances in our understanding of immune dysregulation and therapy for patients. First, they perform deep immune phenotyping on several hundred individuals with DS and demonstrate that immune dysregulation is present from infancy. Second, they report a promising interim analysis of a Phase II clinical trial of a JAK inhibitor in 10 people with DS and moderate to severe skin autoimmunity.Strengths and weaknesses:The relatively large cohort and careful clinical annotation here provide new insights into the immune phenotype of patients with DS. For example, it is interesting that regardless of autoimmune disease or autoantibody status, individuals with DS have elevated cytokines and CRP. Analysis of the cohorts by age demonstrated that some cytokines are significantly elevated in people with DS starting in infancy (e.g., IL-9 and IL-17C). Nearly all adults with DS in this study had autoantibodies (98%) and most had six or more autoantibodies (63%), which differed significantly from euploid study participants. This implies that all patients with DS might benefit from early intervention with therapy to reduce inflammation. However, it is also worth considering that an alternative interpretation that since hypercytokinemia does not vary based on disease state in individuals with DS, this may not be a key factor driving autoimmunity (although it may be relevant for other clinical symptoms such as neuroinflammation).Small case series have suggested the benefit of JAK inhibitors to treat autoimmunity in DS. This is the first report of a prospective clinical trial to test a JAK inhibitor in this setting. The clinical trial entry criteria included moderate to severe autoimmune skin disease in patients aged 12-50 years with DS, and treatment was with the JAK1/3 inhibitor tofacitinib. This clinical trial is a critically important step for the field. The early results support that treatment is well tolerated with an improvement of interferon scores in patients and reduction of autoantibodies. Most patients experienced clinical improvement, with alopecia areata having the greatest response. Treatment may not affect all skin diseases equally, for example of the 5 patients with hidradenitis suppurativa, only 1 showed clinical improvement based on skin score. While very promising, the clinical trial results reported here are preliminary and based on an interim analysis of 10 patients at 16 weeks. Individuals with DS have a lifelong risk of immune dysregulation and thus it is unclear how long therapy, if of benefit, would need to be continued. The results of longer-term therapy will be informative when considering the risks/benefits of this therapy.

We thank the Reviewer for the very positive evaluation. We agree with the Reviewer that the hypercytokinemia of Down syndrome may contribute to other pathophysiological processes beyond autoimmune conditions. Although many cytokines elevated in Down syndrome have well demonstrated pathogenic roles in the etiology of autoimmune diseases in the general population (e.g., TNF-a, IL-6), their consistent upregulation in DS regardless of clinical evidence of autoimmune pathology indicates the existence of a prolonged pre-clinical period, where the hypercytokinemia likely precedes evident tissue damage and symptomology. Alternatively, it is possible that these elevated cytokines are contributing the overall pathophysiology of DS (e.g., neuroinflammation, cognitive impairments, complications from viral infections) without formal diagnosis of an autoimmune disease. We also agree with the Reviewer that not all immune skin conditions would respond equally to JAK inhibition. Based on recent approvals for JAK inhibitors in the immunodermatology field, it is expected that JAK inhibition would show the greatest benefits for alopecia areata, atopic dermatitis, and psoriasis, with less clear results for hidradenitis suppurativa. We hope to contribute to this field through the analysis of the full clinical trial cohort in the near future. Lastly, we strongly agree with the need to assess the value of long-term therapy with JAK inhibitors or other immune therapies in people with Down syndrome for various clinical endpoints.

**Recommendations for the authors:**

**Reviewer #1 (Recommendations For The Authors):**
This paper represents a huge amount of work on a condition whose patients' health and well-being have not always been prioritized, and only relatively recently has the immune dysregulation seen in patients with Down Syndrome (DS) been garnering major research interest.This paper provides an unparalleled examination of immune disorder in patients with DS. In a truly herculean effort, the authors provided the cumulative examination of over 440 patients with DS, confirmed the alterations in immune cell subsets (n=292, 96 controls) and multi-organ autoimmunity seen in these patients as they age, and identified autoantibody production that could contribute to conditions co-occurring in patients with DS. They also sought to look at whether the early immunosenescence seen in DS was due to the inflammatory profile by comparing age-associated markers in DS patients and euploid controls separately, finding that several markers are regulated with age regardless of group, while comparing the effect of age versus DS status on cytokine status identified inflammatory markers elevated in DS patients across the lifespan that do not increase with age or that increase with age only in the DS cohort. This is very interesting in the context of DS in particular, and immunity during aging in general.The second part of the manuscript presents the results from a clinical trial with the JAK inhibitor tofacitinib in DS patients. While the number of DS patients treated with tofacitinib was small, the results were often quite striking. Treatment was well-tolerated and the improvement of dermatological conditions was clear. The less responsive patients AA4 and AA2 provide a very clear illustration that these patients are sensitive to immune triggers during treatment. Additionally, the demonstration that patients' IFN scores and cytokine levels decreased without clear immunosuppression with tofacitinib treatment is encouraging, since treatment with this drug would need to be continuous. I would be curious to see if the patients added past the cutoff for interim analysis follow a similar trajectory. I would not ask the authors to add any data; the paper is well-written and logically constructed.I only have a small comment: I really did not like how Figure 2 a, d, and g tethered the coloring to the magnitude of fold change to show the effect of DS particularly for 2a and 2g. Given that these fold changes are quite modest, the coloring is very light and hard to distinguish. The clear takeaway is that the effect on T cells is greatest, but there must be a better way to illustrate this. Perhaps displaying this graph on a non-white background could help with contrast.

We are grateful for the Reviewer’s very positive assessment of the manuscript and constructive feedback. We want to assure the Reviewer that similar analyses will be completed in the future for the entire cohort recruited into the trial to determine if similar trajectories and results are observed with the larger sample size. Additionally, following Reviewer’s guidance, we have modified the color scales in Figures 2a, d and g so that each panel is on its own dynamic range, thus emphasizing the differences within each immune cell lineage.

**Reviewer #2 (Recommendations For The Authors):**
• Although the focus of the patients in the first part of the paper is on autoimmune/inflammatory conditions, it will be useful to also list the non-autoimmune infectious manifestations for reference with prevalence data. For example, otitis media, or lung infections (mentioned within the paper), or mucosal candidiasis. Same for other manifestations such as cardiac or malignant conditions. Given the impressive number of patients, it will be useful to the readers to have prevalence data for these as well, even in brief statements within the results.

We appreciate this inquiry by the Reviewer. Following Reviewer’s guidance, we have included information on recurrent otitis media, frequent/recurrent pneumonia, congenital heart defects requiring repair, and various forms of leukemia. These additional data are presented in a revised Figure 1 - source data 1 and briefly discussed in the results.

• Have the authors looked at DN T cells and whether they may be enriched in DS patients, given their enrichment in some autoimmune conditions?

Thanks for this inquiry. We did examine DN T cells (double negative T cells), which we referred to in our Figure 2 and Figure 2 – figure supplement 1 as non-CD4+ CD8+ T cells. Although this T cell subset is mildly elevated (in terms of frequency among T cells) in individuals with Down syndrome, the result did not reach statistical significance after multiple hypothesis correction. This negative result is shown in the heatmap in Figure 2 – figure supplement 1d.

• It would be useful to move the segment of the discussion that discusses the interim predefined analysis of the phase 2 trial to the corresponding segment of the results. As this reviewer was reading the paper, it was unclear why the interim analysis was done, whether it was predefined and it was not until the discussion that it became apparent. I believe it will help the readers to have a brief mention that this interim analysis was predefined and set to occur at the first 10 DS enrollees. Also, it would be helpful to state what is the total number of DS patients planned for enrollment in the Phase 2 trial which is continuing recruitment.

We appreciate this comment. Following the Reviewer’s guidance, we have revised the text to explain in the Results section that the interim analysis was predefined and triggered once the first 10 participants completed the 16 weeks of treatment. We also explain that the trial will be considered complete once a total of 40 participants undergo 16 weeks of treatment.

• Although the authors present data on TPO autoantibodies before and after tofacitinib, it remains unclear whether the other non-TPO autoantibodies were altered during treatment or whether this was a TPO autoantibody-specific phenomenon. Was there an alteration in mature B cells or plasmablast populations after tofacitinib? If these data are available, they would further enhance the manuscript. If they are not available, it would be useful for the authors to discuss those in the discussion of the manuscript.

We are grateful for this comment, which strongly aligns with our future research interests and plans for the analysis of the full cohort once the trial is completed. In the interim analysis, we analyzed only auto-antibodies related to autoimmune thyroid disease and celiac disease, as shown in the manuscript. However, we plan to complete a more comprehensive analysis of the effects of JAK inhibition on autoantibody production once the full sample set is available at the end of the trial. Likewise, the clinical trial protocol contemplates collection and processing of blood samples for immune mapping using mass cytometry, which will enable us to answer the question from the Reviewer about potential changes in B cells or plasmablast populations. Following Reviewer’s guidance, we discuss these planned analyses in the Discussion of the revised manuscript.

**Reviewer #3 (Recommendations For The Authors):**
(1) Cellular immune phenotyping data in Figure 2 presents a large number of patients with DS versus euploid controls (292 and 96 respectively). Given the relatively large cohort there would seem to be an opportunity to determine whether age or sex alters the immune phenotype shown, for example, TEMRAs, etc. Was the data analyzed in this way?

We welcome this comment, which clearly aligns with our research interests and planned additional analyses of these datasets generated by the Human Trisome Project. We can share with the Reviewer that although sex as a biological variable has minimal impacts on the strong immune dysregulation observed in Down syndrome, there are clear age-dependent effects, with some immune changes occurring early during childhood versus others taking place later in adult life. A manuscript describing a complete analysis of age-dependent effects on the multi-omics datasets in the Human Trisome Project is currently under preparation.

(2) The authors should strongly consider incorporating/discussing the findings from Gansa et al, Journal of Clinical Immunology May 2024 - where they reviewed the immune phenotype of 1299 patients with Down syndrome.

Thanks for bringing this publication to our attention, which is now cited in the revised manuscript.

(3) It is difficult to differentiate patients Hs2 and Ps1 in Figure 5d.

Thanks for this observation, we have modified the labels for greater clarity in the revised manuscript.

(4) Given their finding of no correlation between cytokine levels/immune phenotype and autoimmunity, some additional discussion of the relevance of hypercytokinemia in the pathogenesis of autoimmunity would seem relevant (given that this was the basis for the clinical trial). The authors mention that cytokine levels may not be appropriate measures of disease in the patients.

We welcome this suggestion and have revised the Discussion along these lines.

(5) Data availability statement: appropriate.